



# Detailed characterization of the CAPS single scattering albedo monitor (CAPS PMssa) as a field-deployable instrument for measuring aerosol light absorption with the extinction-minus-scattering method

Rob L. Modini[1], Joel C. Corbin[2], Benjamin T. Brem[1], Martin Irwin[1*], Michele Bertò[1], Rosaria E. Pileci[1], Prodromos Fetfatzis[3], Kostas Eleftheriadis[3], Bas Henzing[4], Marcel M. Moerman[4], Fengshan Liu[2], Thomas Müller[5], and Martin Gysel-Beer[1]

[1]Laboratory of Atmospheric Chemistry, Paul Scherrer Institute (PSI), 5232 Villigen PSI, Switzerland
[2]Metrology Research Centre, National ResearchCouncil Canada, 1200 Montreal Road, Ottawa K1A 0R6, Canada
[3]Institute of Nuclear and Radiological Science & Technology, Energy & Safety N.C.S.R. "Demokritos", Attiki, Greece
[4]Netherlands Organisation for Applied Scientific Research (TNO), Princetonlaan 6, 3584 Utrecht, the Netherlands
[5]Leibniz Institute for Tropospheric Research (TROPOS), Permoserstrasse 15, 04318, Leipzig,Germany
[*]Now at Catalytic Instruments GmbH, Zellerhornstrasse 7, Rosenheim, 83026, Germany

*Correspondence to*: Rob L. Modini (robin.modini@psi.ch)

**Abstract.** The CAPS PMssa monitor is a recently commercialized instrument designed to measure aerosol single scattering albedo (SSA) with high accuracy (Onasch et al., 2015). The underlying extinction and scattering coefficient measurements made by the instrument also allow calculation of aerosol absorption coefficients via the extinction-minus-scattering (EMS) method. Care must be taken with EMS measurements due to the occurrence of large subtractive error amplification, especially for the predominantly scattering aerosols that are typically found in the ambient atmosphere. Practically this means that although the CAPS PMssa can measure scattering and extinction coefficients with high accuracy (errors on the order of 1 – 10%), the corresponding errors in EMS-derived absorption range from ~10% to greater than 100%. Therefore, we examine the individual error sources in detail with the goal of constraining these as tightly as possible.

Our main focus is on the correction of the scattered light truncation effect (i.e., accounting for the near-forward and -backward scattered light that is undetectable by the instrument), which we show to be the main source of underlying error in atmospheric applications. We introduce a new, modular framework for performing the truncation correction calculation that enables the consideration of additional physical processes such as reflection from the instrument's glass sampling tube, which was neglected in an earlier truncation model. We validate the truncation calculations against comprehensive laboratory measurements. It is demonstrated that the process of glass tube reflection must be considered in the truncation calculation, but that uncertainty still remains regarding the effective length of the optical cavity. Another important source of uncertainty is the cross calibration constant that quantitatively links the scattering coefficient measured by the instrument to its extinction



coefficient. We present measurements of this constant over a period of ~5 months that demonstrate that the uncertainty in this parameter is very well constrained for some instrument units (2 – 3%), but higher for others.

We then use two example field datasets to demonstrate and summarize the potential and the limitations of using the CAPS PMssa for measuring absorption. The first example uses mobile measurements on a highway road to highlight the excellent responsiveness and sensitivity of the instrument, which enables much higher time resolution measurements of relative absorption than is possible with filter-based instruments. The second example from a stationary field site (Cabauw, the

Netherlands) demonstrates how truncation-related uncertainties can lead to large biases in EMS-derived absolute absorption coefficients. Nevertheless, we use a subset of fine-mode dominated aerosols from the dataset to show that under certain conditions and despite the remaining truncation uncertainties, the CAPS PMssa can still provide consistent EMS-derived absorption measurements, even for atmospheric aerosols with high SSA. Finally, we present a detailed list of recommendations for future studies that use the CAPS PMssa to measure absorption with the EMS method. These recommendations could also

be followed to obtain accurate measurements (i.e., errors less than 5 – 10%) of SSA, and scattering and extinction coefficients with the instrument.

## 1. Introduction

Light absorbing aerosols such as black carbon (BC; Bond et al., 2013), brown carbon (BrC; Laskin et al., 2015), tar balls (Corbin and Gysel-Beer, 2019), anthropogenic iron oxide (Moteki et al., 2017), and mineral dust (Sokolik and Toon, 1999)

redistribute radiant energy in the Earth's atmosphere as heat. This perturbs the Earth's radiative balance directly (Haywood and Shine, 1995), and semi-directly through alteration of atmospheric circulation and cloud cover (Koch and Del Genio, 2010). Currently, large discrepancies exist between global climate model simulations of column-integrated aerosol absorption (absorbing aerosol optical depth, AAOD) and sun photometer measurements of the same quantity taken within the AERONET network (Bond et al., 2013; Samset et al., 2018). The uncertainty resulting from this discrepancy feeds into radiative forcing

estimates for absorbing aerosols, contributing to the large and stubborn uncertainty in quantitative estimates of aerosol-radiation climate effects (Myhre et al., 2013). One element that is required to improve this situation and validate both the model simulations and sun photometer measurements is accurate and wide-spread measurements of atmospheric aerosol absorption coefficients ($b_{abs}$). This activity requires sensitive, field-deployable and robust in situ aerosol instrumentation for measuring absorption (Cappa et al., 2016; Lack et al., 2014; Moosmüller et al., 2009).


Traditionally, aerosol light absorption has been measured by detecting the attenuation of light transmitted through aerosol samples deposited on filter substrates (e.g. Rosen et al., 1978). A number of online (i.e., continuously measuring), field-deployable instruments have been developed based on this principle, including the aethalometer (Hansen et al., 1984), the particle soot absorption photometer (PSAP; Bond et al., 1999), and the continuous light absorption photometer (CLAP; Ogren



et al., 2017). An important further development of this class of instruments is the multi-angle absorption photometer (MAAP; Petzold and Schönlinner, 2004), which additionally measures the light backscattered from aerosol-laden filter samples at two separate angles and processes the resulting measurements with a simplified radiative transfer model in order to improve the accuracy of the retrieved aerosol absorption coefficients. Collectively, these instruments are referred to as 'filter-based absorption photometers'.


While the popularity of filter-based absorption photometers has provided critical insights into the optical properties of atmospheric aerosols over the last decades, the limitations of the technique are becoming more problematic as research efforts progress even further. Filter-based light absorption measurements are subject to large positive artefacts due to the effects of multiple scattering from the filter material and the deposited particles, and they are sensitive to aerosol loading, humidity and

aerosol single scattering albedo, SSA (Moosmüller et al., 2009). An additional concern is that the commercial production of some important filter-based instruments has recently been discontinued (e.g. the PSAP by Radiance Research, and the MAAP by Thermo Fisher Scientific).

Motivated by the limitations in the filter-based techniques, instrumentation development efforts have recently focused on

methods for measuring light absorption by aerosols in their natural, suspended state. These techniques include photoacoustic spectroscopy (Arnott et al., 1999; Lack et al., 2006), photo-thermal interferometry (Moosmüller and Arnott, 1996; Sedlacek, 2006), and extinction-minus-scattering (EMS) methods. Here we focus on the EMS method. The EMS method is comprised of two separate underlying measurements: one of the aerosol extinction ($b_{ext}$) and one of the aerosol scattering coefficient ($b_{sca}$). The aerosol absorption coefficient $b_{abs}$ is then obtained by subtracting $b_{sca}$ from $b_{ext}$:


$$b_{abs} = b_{ext} - b_{sca} \qquad . \tag{1}$$

Traditionally, EMS measurements have been performed by two separate instruments (e.g. an integrating nephelometer for aerosol scattering, and a separate extinction monitor). Additionally, the use of EMS measurements has mostly been limited to the laboratory where high absorption signals are easily achievable, and artefacts (e.g., due to the scattered light truncation

effect) can be avoided. In such a laboratory setting, EMS measurements are considered as a primary standard for measuring aerosol absorption thanks to the traceability of the underlying $b_{ext}$ and $b_{sca}$ measurements (e.g. Bond et al., 1999; Schnaiter et al., 2003; Virkkula et al., 2005).

The continued development of sensitive techniques for measuring $b_{ext}$ using multi-pass optical cavities (e.g. cavity ring-down

spectroscopy, Moosmüller et al., 2005; and cavity attenuated phase shift spectroscopy, CAPS, Kebabian et al., 2007) has created the possibility of extending application of the EMS technique more broadly to different types of atmospheric and/or


test bench (i.e., emissions) measurements. This endeavor poses several challenges: i) subtractive error amplification in EMS-derived $b_{abs}$ can become very large when $b_{sca}$ is close to $b_{ext}$ (i.e., as SSA → 1), which occurs very commonly throughout the Earth's atmosphere (Dubovik et al., 2002), ii) artefacts such as the scattered light truncation effect in integrating nephelometer

measurements of $b_{sca}$ are generally unavoidable and more difficult to quantify for ambient aerosols (which are typically complex mixtures of particles of varying size, composition and morphology), iii) it is more difficult to ensure thorough and regular instrument calibrations in a field versus a laboratory setting, and iv) it is usually more difficult to control sampling arrangements in the field to ensure that $b_{ext}$ and $b_{sca}$ are measured under the same (or at least well-known) environmental conditions.


Despite these challenges, the possibility of performing EMS measurements of atmospheric aerosol absorption has recently been boosted by the development and commercialization of the cavity attenuated phase shift SSA monitor (CAPS PMssa) by Aerodyne Research Inc. (Billerica, MA, USA; Onasch et al., 2015). The CAPS PMssa monitor combines measurements of $b_{ext}$ and $b_{sca}$ in a single instrument and sample volume, following in the tradition of earlier combined extinction-scattering

instruments (Gerber, 1979; Sanford et al., 2008; Strawa et al., 2003; Thompson et al., 2008). Its direct precursor instrument – the Aerodyne CAPS extinction monitor (CAPS PMex) – uses the CAPS technique to measure $b_{ext}$ values with high sensitivity in a compact optical cavity and overall instrument unit (Massoli et al., 2010; Petzold et al., 2013). The CAPS PMssa is based on the same optical cavity, but additionally includes an integrating sphere reciprocal nephelometer around the cavity for measurement of $b_{sca}$.


The CAPS PMssa was originally designed to measure SSA (i.e., the ratio of $b_{sca}$ to $b_{ext}$), a quantity which is not subject to the same subtractive errors as $b_{abs}$. However, its design addresses two of the key challenges of atmospheric EMS measurements that were listed in the paragraph above, which makes it an attractive candidate for performing such measurements. Specifically, by simultaneously measuring $b_{ext}$ and $b_{sca}$ for the same volume of air, there is no need to account for possible differences in

environmental conditions or sampling losses that could affect these two coefficients. Additionally, this feature allows the cross calibration of one coefficient against the other using white test aerosols (non-absorbing, i.e., where $b_{ext} = b_{sca}$), which facilitates the development of relatively simple field calibration procedures (in practice, $b_{sca}$ is cross calibrated against $b_{ext}$ in the CAPS PMssa). Nevertheless, great care must still be taken when performing EMS measurements with the CAPS PMssa to ensure that errors in the underlying $b_{ext}$ and $b_{sca}$ measurements are minimized and that very large subtractive error amplification is

avoided. This essentially reduces down to the following problem: errors that may be acceptable if one is interested in measuring $b_{ext}$, $b_{sca}$, or SSA (say on the order of 5 – 10%), are substantially magnified – perhaps to over 100%, as we will show below – when using the very same measurements to derive $b_{abs}$. Therefore, the user must be concerned about errors on the order of only a few percent if they wish to use the CAPS PMssa to reliably measure atmospheric aerosol absorption coefficients.



One of the key sources of uncertainty that must be considered for the CAPS PMssa (and integrating nephelometry in general)
is the scattered light truncation effect (e.g. Moosmüller and Arnott, 2003; Varma et al., 2003). Integrating nephelometers seek
to detect light scattered in all possible directions. In reality, a fraction of near-forward and near-backward scattered light is
always lost due to unavoidable physical design limitations. As a result, $b_{sca}$ measurements are biased low and need to be
corrected. The required correction factor depends on a particular instrument's geometry, as well as the angular distribution of

light scattered from an aerosol sample, which is a function of the optical wavelength, and the size distribution, composition,
mixing state, and morphology of the particles in that sample.

Onasch et al. (2015) presented a simple model for calculating truncation correction factors for the CAPS PMssa based on Mie
theory calculations with inputted particle size distributions. However, this model does not consider an important physical

process that serves to increase scattered light truncation: reflection of scattered light from the inner surface of the glass
sampling tube within the integrating nephelometer. Liu et al. (2018) developed a more sophisticated truncation model based
on solution of the radiative transfer equation (RTE) configured specifically to the PMssa optical system. As well as allowing
for the treatment of non-spherical particles (which is not possible with Mie theory), the RTE approach also allows for the
treatment of additional physical processes (e.g. multiple scattering from the aerosol and glass tube reflection). CAPS PMssa

truncation values calculated with these models have so far been validated against only a limited dataset of experimental
measurements (Onasch et al., 2015). Furthermore, there is a lack of systematic analyses that aim to determine the sensitivity
of EMS-derived $b_{abs}$ values to changes in calculated truncation (e.g. for ambient aerosol samples).

Despite the many unresolved uncertainties, the CAPS PMssa has already been used as an instrument for measuring $b_{abs}$ in a

number of different ambient field campaigns (Chen et al., 2018; Han et al., 2017; Xie et al., 2018), emissions testing
experiments (Corbin et al., 2018; Zhai et al., 2017), and soot characterization experiments (Dastanpour et al., 2017; Forestieri
et al., 2018; Perim de Faria et al., 2019).

In this study, we present a compilation of theoretical calculations, novel laboratory measurements, and example field

applications that all serve a common purpose: to improve the truncation correction approach and to determine the extent to
which the CAPS PMssa can be used to measure aerosol absorption coefficients via the EMS method.

In Sect. 2 we present a theoretical description of the instrument, including the introduction of a new truncation model that
includes the process of glass tube reflection and is suitable for application to large field datasets. This Section culminates in

the presentation of a detailed $b_{abs}$ error model, which is used to demonstrate why it is so critical to constrain errors in the
truncation calculations and instrument cross calibration constant. This finding motivates the experimental work described in
the remainder of the paper. Section 3 details the experimental methods used. Section 4 then presents some regular
measurements of the CAPS PMssa cross calibration constant in order to assess its precision and stability. In Sect. 5 we compare



the results of novel and comprehensive laboratory truncation measurements with calculated values from a range of different

truncation models. Synthesizing all of these issues together, Sect. 6 then presents two example field datasets that demonstrate both the potential and the limitations of using the CAPS PMssa to measure atmospheric aerosol absorption. Finally, in the concluding Sect. 7 we present a list of recommendations for future CAPS PMssa studies.

**2. Theoretical description of the CAPS PMssa monitor**

**2.1 General introduction**

The CAPS PMssa monitor is described in detail previously in the original technical paper by Onasch et al. (2015). A schematic diagram of the instrument is shown in Fig. 1. Briefly, the instrument consists of an optical cavity formed by two high reflectivity mirrors (reflectivity ~ 0.9998), creating a long effective optical path length (~ 1 – 2 km). Aerosol samples are drawn continuously through this cavity at a flow rate of 0.85 litres per minute (light blue arrows in Fig. 1) with no size selection performed at the instrument inlet, meaning that the samples generally contain both sub- and supermicrometer particles.

Smaller, particle-free purge flows of ~0.025 litres per minute are drawn continuously over the high reflectivity mirrors to prevent their contamination (green arrows in Fig. 1). The purge and sample flows are generated from the same double-headed membrane pump.

The input light source to the cavity is provided by a single light emitting diode (LED). Units are available from the

manufacturer Aerodyne Research, Inc. with LEDs centred at wavelengths of 450, 530, 630, 660 and 780 nm. The intensity of the LED input light is square-wave modulated (typically at 17 kHZ), and the intensity of light leaking through one mirror is monitored by a vacuum photodiode. The intensity of the light circulating in the cavity increases exponentially during the LED on-phase and decreases exponentially during the LED off-phase, with a timescale dependent on the reflectivity of the mirrors and optical loss in the cell (Lewis et al., 2004). The introduction of a scattering or absorbing species to the cell enhances this

optical loss, resulting in a shorter optical lifetime in the cavity and a phase shift of the output signal relative to the input signal. This phase shift is measured by the vacuum photodiode using a quadrature signal integration method (Kebabian et al., 2007). This is the technique for measuring extinction coefficients known as cavity attenuated phase shift spectroscopy (CAPS) and its application in the CAPS PMssa is referred to as the 'extinction channel' of the instrument.

The second light detector in the instrument is a photomultiplier tube (PMT) that is used to measure the integrated aerosol scattering coefficient (Fig. 1). It is referred to as the 'scattering channel' of the instrument. The PMT is placed on the integrating sphere that surrounds the centre of the optical cavity. The integrating sphere has an inner diameter of 10 cm. The inside of the integrating sphere is coated white to form a Lambertian reflector (reflectivity = 0.98), which functions to maximize the amount of scattered light detected by the PMT and to minimize any bias between light collected from different scattering angles.

Onasch et al. (2015) calculated that the variation in the angular sensitivity of the sphere as a function of scattering angle is less

than 1%. The integrating sphere does not contain a baffle as described by Onasch et al. (2015). A glass tube with inner diameter of 1 cm passes through the centre of the integrating sphere in order to encapsulate the aerosol flow along the central axis of the optical cavity.

In this study we define the central axis of the optical cavity as the $z$-dimension and the centre of the integrating sphere to be at position $z = 0$ cm. A particle lying along the central $z$-axis scatters light in polar directions at scattering angles $\theta$ defined with respect to the z-axis (two limiting examples for forward- ($\theta_1$) and back-scattered ($\theta_2$) light are shown in Fig. 1), and azimuthal directions at scattering angles $\varphi$ (not shown in Fig. 1).

## 205 2.2 Data processing and important correction and calibration factors

The data processing chain applied by the CAPS PMssa instrument firmware to calculate aerosol extinction and scattering coefficients from the measured photodiode and PMT signals is displayed in Fig. 2 (Onasch et al., 2015). The instrument has two modes of operation where data are collected: sample and baseline measurements. The sample and baseline measurements are achieved by a controlled three-way valve that directs the sampled air either directly into the optical cavity or first through 210 a filter that removes all particles. The instrument firmware allows the baseline measurements to be repeated automatically at a frequency and duration set by the user. Typically during field operation baseline measurements are performed for 1 minute every 5 or 10 minutes.

In the extinction channel, the sample and baseline measurements are first treated by subtracting out a constant factor that 215 accounts for extinction due to Rayleigh light scattering from the aerosol carrier gas. The subtraction term is corrected using temperature and pressure measurements taken by the instrument to account for possible variations in these quantities between sample and baseline periods.

Full treatment of the PMT scattering signals is given by Onasch et al. (2015). The scattering signals are counted during the 220 LED off-phase when only highly collimated light is circulating in the cavity in order to minimize the contribution of light scattered from interior surfaces of the instrument. Consequently, the average intensity of circulating light during the LED off-phase must be accounted for in the scattering calculation, as illustrated by the dot-dashed lines in Fig. 2 and described in detail in Onasch et al. (2015).

Following these initial data treatment steps, uncorrected aerosol extinction and uncalibrated scattering coefficients ($b_{ext,uncorr.}$ and $b_{sca,uncalib.}$, respectively) are obtained by taking the difference between the sample-mode coefficient measurements (which we term $b_{ext,sample}$ and $b_{sca,sample}$) and the interpolated baseline-mode coefficient measurements ($b_{ext,baseline}$ and


$b_{sca,baseline}$). By default, the instrument firmware uses a step function to interpolate the baseline values between each baseline period (i.e., the mean value of a baseline period is assumed to stay constant until it is replaced by the mean value of the next

baseline period). However, the data output files from the instrument also provide sufficient information for the user to apply custom methods for calculating the interpolated coefficients $b_{ext,baseline}$ and $b_{sca,baseline}$ (e.g. linear or cubic spline interpolation; Pfeifer et al., 2020).

Following the sample-baseline difference calculations, one extinction correction factor (the geometry correction factor, $\alpha$) and

two scattering correction factors (cross calibration, $\beta$, and truncation factors, $\gamma$) are multiplicatively applied to the respective signals in order to obtain the calibrated and corrected aerosol coefficients $b_{ext}$ and $b_{sca}$. The $\alpha$ and $\beta$ factors are applied automatically by the instrument firmware, while $\gamma$ must be applied manually by the user in post-processing. All three correction factors are discussed in detail in the Sections below. The aerosol absorption coefficient is then obtained as:

$$b_{abs} = b_{ext} - b_{sca} = \frac{1}{\alpha} \cdot \left( b_{ext,sample} - b_{ext,baseline} \right) - \frac{\gamma}{\beta} \cdot \left( b_{sca,sample} - b_{sca,baseline} \right) \qquad . \tag{2}$$


### 2.2.1 Geometry correction factor ($\alpha$)

The purge flows that protect the high reflectivity mirrors in the CAPS PMssa shorten the effective optical path length of the cavity and may slightly dilute the instrument sample flow at the cavity inlet. Therefore, a correction factor must be applied to the measured extinction coefficients in order to account for these changes (Massoli et al., 2010; Onasch et al., 2015; Petzold

et al., 2013). We refer to this correction factor as the geometry correction factor, $\alpha$, which can be determined by external calibration, i.e., by comparing CAPS PMssa measurements against independently measured or calculated $b_{ext}$ (e.g. Mie-calculated $b_{ext}$ values for spherical, monodisperse test aerosols, Petzold et al., 2013; or measured $b_{ext}$ values for non-absorbing test aerosols obtained with a reference nephelometer, Pfeifer et al., 2020).

Onasch et al. (2015) applied the Mie calculation approach to measurements of polystyrene latex (PSL) spheres of varying diameter to determine an $\alpha$ value of 0.73 for a CAPS PMssa unit operating at 630 nm. This is lower than the general value of 0.79 quoted by Onasch et al. (2015) for CAPS PMex monitors, which they note is expected due to small differences in the cavity geometries. The CAPS PMssa units used in this study (Table 2) participated in European Center for Aerosol Calibration (ECAC; http://www.actris-ecac.eu/) workshops (CAPS630b in August 2016; and CAPS450, CAPS630a, CAPS780 in January

2017) where their geometry correction factors were determined against reference instrumentation (CAPS PMex, nephelometer) using ammonium sulphate test aerosols. The units were determined to have $\alpha$ values of 0.78 (CAPS450), 0.71 (CAPS630a), 0.7 to 0.73 (CAPS630b), and 0.78 (CAPS780). Therefore, it appears that $\alpha$ is instrument-unit-dependent. The stability of $\alpha$ over time is still an open question. However, regular and frequent measurements of $\alpha$ in CAPS PMex monitors performed at



the ECAC suggests that it does not drift by more than 3% over the period of a year. By default, the CAPS PMssa firmware
automatically applies an $\alpha$ factor of 0.7 to calculate $b_{ext}$ (Fig. 2).

**2.2.2 Scattering cross calibration factor ($\beta$)**

The scattering cross calibration factor ($\beta$) is used to relate the PMT-measured scattering signal of the CAPS PMssa to an
absolute aerosol scattering coefficient. The value of $\beta$ can be determined by cross calibrating the uncalibrated aerosol scattering
coefficient $b_{sca,uncalib.}$ against $b_{ext}$ measured by the extinction channel (Onasch et al., 2015). This approach is possible because
the scattering and extinction coefficients are measured simultaneously for the same air sample, and $b_{ext}$ measured using the
CAPS method is effectively 'calibration free' (apart from the geometry correction factor, as discussed in Sect. 2.2.1, as well
as potential non-linearities at high baseline losses). Amongst other factors, $\beta$ depends on the PMT detector response, which
can vary over time. Therefore, regular cross calibrations should be performed.

Non-absorbing test samples are required to perform the cross calibration and to determine a value for $\beta$ (i.e., purely scattering
samples for which $b_{ext} = b_{sca}$, or SSA = 1). In principle, the calibration can be performed with gases or aerosol particles. In
practice, we performed all calibrations in the present study with particles because readily available calibration gases such as
$N_2$ and $CO_2$ span a much smaller range in $b_{sca}$ than is achievable with aerosols of different concentrations, additional corrections
are required to account for the changes in optical path length and dilution with the purge flows for different gases (see Sect.
2.2.1), and we have observed that the instrument can take a long time (~hours) to adjust and stabilize when filled with different
gases.

When using the particle-based calibration method, non-absorbing aerosol particles with size parameters $x$ in the Rayleigh light
scattering regime should be used to ensure well-defined scattered light truncation, since the scattering phase function is
independent of particle size in the Rayleigh regime. We term cross calibration constants derived in this specific manner as
$\beta_{Rayleigh}$. The size parameter $x$ relates the aerosol particle diameter $D_p$ to the wavelength of light $\lambda$ through the expression $\pi D_p / \lambda$.
The Rayleigh regime is defined by the condition $x \ll 1$. In practice, there is a trade-off between selecting particle sizes that are
small enough to lie within or near the Rayleigh regime limit but large enough to generate scattering and extinction signals with
sufficiently high signal-to-noise ratios. This means particles with diameter less than approximately 150 nm should be used to
determine $\beta_{Rayleigh}$ in the 450 nm CAPS PMssa, while slightly larger particles (e.g. $D_p \sim 200$ nm) can be used with 630 or 780
nm CAPS PMssa instruments.

Formally, the Rayleigh-regime, particle-based cross calibration approach can be expressed as:



$$\beta_{Rayleigh} = \frac{b_{sca,uncalib.}^{non-abs,Rayleigh}}{b_{ext}^{non-abs,Rayleigh}} = \alpha \cdot \left( \frac{b_{sca,uncalib.}^{non-abs,Rayleigh}}{b_{ext,uncorr.}^{non-abs,Rayleigh}} \right) = \alpha \beta'_{Rayleigh} \quad , \tag{3}$$


where $b_{ext}^{non-abs,Rayleigh}$ and $b_{sca,uncalib.}^{non-abs,Rayleigh}$ are the extinction and uncalibrated scattering coefficients, respectively, for a population of non-absorbing particles with size parameters in the Rayleigh regime. The right-hand side of Eq. (3) is obtained by substitution of the relationship $b_{ext} = \frac{1}{\alpha} b_{ext,uncorr.}$ into the left-hand side ratio. From this substitution, it can be seen that $\beta_{Rayleigh}$ (and $\beta$, generally) is directly proportional to the geometry correction factor $\alpha$, which is required to measure $b_{ext}$

accurately as discussed in Sect. 2.2.1 (the remaining fraction of the cross calibration constant is termed $\beta'_{Rayleigh}$ to distinguish it from $\beta_{Rayleigh}$). Thus, Eq. (3) demonstrates how the cross calibration approach quantitatively links $b_{sca}$ to $b_{ext}$ in the CAPS PMssa. Following application of $\beta_{Rayleigh}$, we refer the calibrated aerosol scattering coefficient corrected for the truncation of Rayleigh scattered light as $b_{sca,Rayleigh}$. This is to recognize the fact that the cross calibration approach represented by Eq. (3) implicitly corrects for the truncation of light scattered from the calibration aerosol, which has been chosen specifically to have

the well-defined phase function corresponding to Rayleigh light scattering.

Onasch et al. (2015), demonstrated that the linearity shown by Eq. (3) is valid up to extinction coefficients of ~1000 Mm$^{-1}$, which is higher than typical ambient aerosol extinction coefficients, excluding perhaps coefficients in heavily polluted urban environments. The precise limit of linearity should be examined for individual instrument units if it is relevant for a particular

experiment. For very high aerosol loadings above the limit of linearity the CAPS PMssa cross calibration approach can still be used. However, this requires the addition of empirically-derived higher order terms in $b_{ext}^{non-abs,Rayleigh}$ to Eq. (3). In addition, the potential occurrence of multiple scattering effects needs to be considered at very high aerosol loadings (Wind and Szymanski, 2002).

### 2.2.3 Truncation correction factor ($\gamma$)

The final quantitative correction factor that must be applied to the scattering coefficients measured with the CAPS PMssa is the truncation correction factor, $\gamma$. The truncation correction factor $\gamma$ is applied to $b_{sca,Rayleigh}$ to compensate for the light scattered in near-forward and near-backward directions that is not measured by the instrument due to geometric restrictions. The truncation correction factor $\gamma$ depends on both the instrument properties as well as the angular distribution of light scattered from the aerosol sample being measured (referred to in short as the ensemble scattering phase function, $S_p$), which depends on

the aerosol size distribution, morphology, mixing state, and composition (refractive indices). The existing methods for calculating the CAPS PMssa truncation correction factor $\gamma$ either do not include the process of scattered light reflection from the inner surface of glass sampling tube (Onasch et al., 2015), or are computationally expensive (Liu et al., 2018) and not well-





suited to calculate time-resolved truncation factors for large datasets (e.g. as required for the example Cabauw dataset in Sect. 6.2). Therefore, we present here a new truncation calculation framework that overcomes both of these limitations.


The new calculation framework is presented visually as a flow chart in Fig. 3. The full set of details and equations is given in Appendix 1. Briefly, we define $\gamma$ as the normalized ratio of the true integrated scattering coefficient, $b_{sca,true}$ to the truncation-affected scattering coefficient that is actually accessible to measurement, $b_{sca,meas}$. The true scattering coefficient $b_{sca,true}$ represents the coefficient that would be measured by an ideal integrating nephelometer capable of collecting light scattered in all possible directions. The ratio requires normalization by a factor $k_{Rayleigh}$ to represent the fact that some scattered light truncation is already included implicitly in the cross calibration constant, due to the way in which it is measured. For the recommended case of cross calibration with Rayleigh scatterers according Eq. (3), $k_{Rayleigh}$ represents the truncation of the Rayleigh scattered light from the calibration aerosol. That is,

$$\gamma = \frac{b_{sca,true}}{b_{sca,meas}} \cdot k_{Rayleigh} = \frac{b_{sca,true}}{b_{sca,meas}} \cdot \left( \frac{b_{sca,meas}^{Rayleigh}}{b_{sca,true}^{Rayleigh}} \right) \ . \tag{4}$$


Defined in this manner, $\gamma$ equals 1 for aerosols in the Rayleigh regime. For aerosols containing larger particles or non-spherical particles that produce more forward-focused light scattering, $\gamma$ is always greater than 1.

The equations for calculating the integrated scattering coefficients in Eq. (4) are detailed in Appendix A1. These equations

have been given in several previous publications (Anderson et al., 1996; Heintzenberg and Charlson, 1996; Moosmüller and Arnott, 2003; Müller et al., 2011b; Peñaloza M, 1999). The novel aspect of our formulation is that we explicitly define a function representing the efficiency with which an integrating nephelometer is able to collect scattered light, $\eta(\theta, \lambda)$, which is a simple function varying between 0 and 1. Values of 0 indicate that a nephelometer collects no light of wavelength $\lambda$ at some scattering angle $\theta$, while values of 1 indicate that a nephelometer collects all the light scattered at angle $\theta$. Considering $\eta(\theta, \lambda)$

explicitly has a number of advantages: i) it allows transparent representation of an instrument's truncation angles (i.e., by setting $\eta$ equal to 1 between two truncation angles, and 0 beyond them), (ii) it allows for the simple and explicit introduction of additional physical processes into light scattering calculations (e.g. reflection from the glass sampling tube can be considered by combining the Fresnel equation for reflection probability with $\eta$, as shown in Appendix A1), (iii) it provides a clear and intuitive way to compare the abilities of different nephelometers to collect scattered light, and (iv) it emphasizes the modular

nature of the truncation calculation.

One of the important characteristics of integrating sphere type reciprocal nephelometers like the CAPS PMssa is that truncation is a function of position along the central axis of the optical cavity (which we denote as the $z$-dimension, Fig. 1). This



characteristic is represented by the small subplots in Fig. 3 that show light collection efficiency curves (termed $\eta\_spot$ in

Appendix A1) for six different $z$ positions in the CAPS PMssa, including for two positions at 1 cm outside of the integrating sphere (i.e., $z = -6$ and 6 cm). Positions outside of the integrating sphere must be considered since it is possible for particles outside the sphere to scatter light into the sphere (e.g. Varma et al., 2003), even if only through a narrow range of scattering angles. We term the extra length that needs to be considered outside the sphere's boundaries as the $l$ parameter. The geometrical limits for the $l$ parameter are 0 (i.e., no extra path length considered) and 4.7 cm (the distance between the integrating sphere

and the sample inlet and outlet ports to the optical cavity). Onasch et al. (2015) and Liu et al. (2018) both used $l = 1$ cm in their calculations (i.e., they considered a $z$-range from -6 to 6 cm). The $\eta\_spot$ subplots in Fig. 3 also demonstrate the effect of glass tube reflection: between a sphere's truncation angles, reflection decreases the probability of light collection from 1 to some value less than 1. Therefore, glass tube inner surface reflection serves to increase scattered light truncation. A single, integrated light collection efficiency function for the CAPS PMssa can be generated by integrating $\eta\_spot$ over all possible $z$ positions

(Eq. A13). CAPS PMssa integrated $\eta$ functions are shown in Fig. 3 for the two cases of without and with glass reflection.

It is important to stress the implications of the modularity of truncation calculation. This modularity means that once the $\eta(\theta, \lambda)$ and angular sensitivity functions are known for a particular instrument, they can be combined with any measured or calculated ensemble scattering phase function in order to calculate $\gamma$. In the present study, we used Mie theory and co-located

particle size distribution measurements to efficiently calculate hourly-resolved $S_p$ functions and $\gamma$ values for a month-long field campaign (Sect. 6.2; Fig. S12). This Mie calculation method assumes spherical, homogeneous particles. If one wished to consider more complex particle morphologies, a more sophisticated optical model could be used to calculate the scattering phase functions $S_p$. Or, if co-located polar nephelometer measurements of the scattering phase function were available (e.g. Espinosa et al., 2018), these could be input directly into the truncation calculation.

**2.3 Absorption error model for the CAPS PMssa and discussion of the sources and effects of uncertainties in $\beta$ and $\gamma$**

It is critical to carefully consider and understand the sources of errors in EMS-derived $b_{abs}$ values, since these can be very large when taking the difference of two potentially larger numbers – $b_{sca}$ and $b_{ext}$ – that each carry their own uncertainties. Based on the data processing framework presented in the previous Sect. 2.2, an error model can be constructed for CAPS PMssa absorption coefficients by considering the uncertainty in each of the individual parameters in the right-hand side of Eq. (2) and

applying the standard rules of error propagation, including consideration of potential covariance of the errors in $b_{sca}$ and $b_{ext}$. The explicit equations for such a model are given in Appendix A2. Table 1 lists the individual parameters in the error model along with realistic estimates of their uncertainties. In general, we consider two sources of uncertainties: uncertainty due to the limited precision with which a particular parameter can be determined during calibration or measurement, and uncertainty due to possible drift of a parameter between available calibrations or measurements (e.g. baseline drift between two subsequent

baseline measurements). For a given parameter, these two sources of errors are independent and they can be added in



quadrature, or if one of the errors is much larger than the other, this larger error can simply be used in error propagation calculations.

Many of the individual uncertainty estimates given in Table 1 are taken from previous studies and will not be discussed in
great detail here. However, the uncertainties in the $b_{sca}$ correction factors $\gamma$ and $\beta$ are still poorly constrained and require further investigation. We refer to these uncertainties as $\delta\gamma$ and $\delta\beta$, respectively. Onasch et al. (2015) showed that $\beta$ can be measured with high precision for a 630 nm PMssa unit, but the obtainable precision at other operation wavelengths as well as the stability in $\beta$ over time have not been fully explored. Therefore, the overall $\delta\beta$ is still not well characterized.

The uncertainty in $\gamma$ is more difficult to quantify. At the highest level it can be categorized into uncertainties related to the instrument properties (e.g. should glass tube reflection be considered, and an appropriate $l$ value), and those related to knowledge of the scattering phase functions of the aerosol samples being measured. Regarding uncertainties in the latter category, these can be further characterized depending on how the angularly resolved light scattering information is obtained. In the best case scenario, the scattering phase functions would be obtained directly from co-located polar nephelometer
measurements, in which case $\delta\gamma$ would depend on the accuracy of these measurements (and possible extrapolation of those measurements beyond a polar nephelometer's truncation angles). Since polar nephelometer measurements are rarely performed in measurement campaigns, it is more likely that scattering phase functions will be calculated with an optical model (e.g. Mie theory) using co-located size distribution measurements (covering both sub- and supermicrometer size fractions) as input. In this case, $\delta\gamma$ will be a function of the accuracy of the input size distribution measurements, as well as the representativeness of
the optical model and its inputs (e.g. complex refractive index, particle morphology if the optical model includes treatment of this). In the worst case scenario, which is expected to occur frequently in field work, there might be no information available to constrain the scattering phase function. In this case, $\gamma$ values would need to be assumed. For example, a user might simply assume that $\gamma$ equals 1, which is equivalent to assuming that all particles in the sample are Rayleigh light scatterers. In this case, $\delta\gamma$ should reflect the possible consequences of that assumption. In Table 1 we provide some estimates for both $\delta\gamma$ and $\delta\beta$
that are based on the results of the present study. These estimates and results are discussed in specific detail below in Sections 4, 5, and 6.

For now, we use our error model to assess the possible impacts of $\delta\gamma$ and $\delta\beta$ on the relative uncertainty in EMS-derived $b_{abs}$, regardless of where the uncertainty in these two parameters actually comes from. Indeed, we generalize this analysis even
further by considering the relative uncertainty in the combined $b_{sca}$ correction factor $\gamma/\beta$, given by the equation:

$$\frac{\delta\left(\gamma/\beta\right)}{\gamma/\beta} = \sqrt{\left(\frac{\delta\gamma}{\gamma}\right)^2 + \left(\frac{\delta\beta}{\beta}\right)^2} \qquad . \tag{6}$$



This approach is motivated by the fact that $\delta\gamma$ and $\delta\beta$ have equal impacts on the uncertainty in EMS-derived $b_{abs}$, and it is justified because $\delta\gamma$ and $\delta\beta$ are independent of one another.


The relative uncertainty in $b_{abs}$ calculated with our error model can be interpreted as the precision with which $b_{abs}$ can theoretically be determined for a given set of error model inputs. It should be noted that in addition to this precision-based uncertainty, the absolute accuracy of $b_{abs}$ will also depend directly on the accuracy of the geometry correction factor $\alpha$ if the instrument is cross calibrated as recommended in Sect. 2.2.2. This is because in the same manner as with $b_{sca}$, the cross

calibration serves to define $b_{abs}$ with respect to $\alpha$, which can be seen by substituting the right-hand side of Eq. (3) into Eq. (2):

$$b_{abs} = \frac{1}{\alpha} \cdot \left( b_{ext,uncalib.} - \frac{\gamma}{\beta'_{Rayleigh}} \cdot b_{sca,uncalib.} \right) \quad . \tag{7}$$

In the present study we do not explicitly consider the $\alpha$-related uncertainty in $b_{abs}$, though it is important to keep this in mind.
Specifically we note that the errors in $\alpha$ cause covariant errors in $b_{ext}$ and $b_{sca}$. Hence, the relative error in $\alpha$ propagates 1-to-1 to corresponding relative error in EMS-derived $b_{abs}$, independently of SSA. This is not the case for errors in $\beta'_{Rayleigh}$, for example, which lead to error in $b_{sca}$ that is independent of errors in $b_{ext}$, and therefore relative errors in $b_{abs}$ that do vary with SSA. In practice, the uncertainty due to $\alpha$ can only be determined by comparison of CAPS PMssa measurements against an independent reference. It is also worthwhile to note that the uncertainty in SSA measured by CAPS PMssa does not depend
on the uncertainty in $\alpha$, since this factor simply cancels out when taking the ratio of $b_{sca}$ to $b_{ext}$. This is one of the key design features of the cross-calibrated instrument (i.e., the relative error in $\alpha$ makes identical and covariant contributions to the errors in $b_{ext}$, $b_{sca}$, and $b_{abs}$).

Focusing on the precision-related uncertainty in $b_{abs}$ that is quantified by our error model (Eq. A16), Fig. 4 displays this variable
as a function of the combined relative uncertainty in $\beta$ and $\gamma$ for a range of different atmospheric conditions (two different aerosol loadings and 4 different SSA values). The curves in this figure were generated using the following model inputs designed to represent the CAPS630b instrument characteristics during the Cabauw field campaign (Sect. 6.2): [$\alpha$=0.7, $\delta\alpha$=0, $\beta$=0.81, $\gamma$=1.04, $\delta b_{ext,sample}$= $\delta b_{sca,sample}$=1/$\sqrt{3600}$ Mm$^{-1}$, $b_{ext,baseline}$=512 Mm$^{-1}$, $\delta b_{ext,baseline}$=0.35 Mm$^{-1}$, $b_{sca,baseline}$=50 Mm$^{-1}$, $\delta b_{sca,baseline}$=0.66 Mm$^{-1}$]. Parameter $\delta\alpha$ was set to 0 to reflect the fact that the accuracy of $\alpha$ is not considered in the simulation,
as well as the assumption that $\alpha$ does not vary between subsequent cross calibration measurements. Fig. 4 can be interpreted as follows: taking an uncertainty of 5% for $\gamma$ and 2% for $\beta$ (which we will show later to be realistic estimates), the relative



uncertainty in the combined $b_{sca}$ correction factor equals 5.4% based on Eq. (6). This example corresponds to vertical blue dashed line in Fig. 4. Two other realistic examples are also shown in the Figure as vertical dashed lines.

Several important and general features are apparent in Fig. 4. Firstly, it is seen that the precision-related uncertainty in $b_{abs}$ increases dramatically with small increases in uncertainty in either $\beta$ or $\gamma$. As a result, small uncertainties in $\beta$ or $\gamma$ can result in large uncertainties in $b_{abs}$. The relative uncertainty in $b_{abs}$ is also a strong function of SSA due to the large subtractive error amplification that results from taking the difference of two large and uncertain numbers. Taking these two points together and considering the example case demonstrated by the vertical red dashed line, a combined uncertainty of only 10.2% in $\gamma$ and $\beta$

leads to precision-related uncertainties in $b_{abs}$ of over 80% at SSA greater than 0.9. Such large SSA is very common for atmospheric aerosols, which highlights why it is so critical to minimize uncertainties in $\beta$ and $\gamma$ when using the CAPS PMssa to measure atmospheric aerosol absorption with the EMS method.

The divergences between the corresponding dashed and solid grey lines in Fig. 4 represent the effects of the errors in both the

extinction and scattering baseline signals. These errors can be important under very clean atmospheric conditions (represented by the case $b_{ext} = 10$ Mm$^{-1}$), since the absolute differences between sample-mode and baseline signals are then small. However, these sources of uncertainty are quickly overwhelmed as uncertainties in $\beta$ and $\gamma$ increase, resulting in the convergence of the pairs of dashed and solid grey lines moving from left to right across the figure. For the high aerosol load case (represented by $b_{ext} = 100$ Mm$^{-1}$), it is interesting to note that for 0% uncertainty in $\beta$ and $\gamma$, the relative uncertainty in $b_{abs}$ is still SSA dependent,

even though $b_{sca}$ has been defined with respect to $b_{ext}$ by the cross calibration and $\delta\alpha$ set to 0 in the simulation. This is because $b_{ext,sample}$ and $b_{sca,sample}$ still carry independent uncertainty due to random noise, even if this is relatively small (i.e., 1 Mm$^{-1}$ at 1 sec temporal resolution).

The $b_{abs}$ uncertainty values displayed in Fig. 4 were simulated to represent 1-hour averaged measurements. Fig. S1 indicates

that the equivalent values representing 1-minute averaged measurements are practically equivalent to those shown in Fig. 4, while those representing 1-second measurements are only greater for low values of uncertainty in $\beta$ and $\gamma$. This is because of all the uncertainties listed in Table 1, only the uncertainties in $b_{ext,sample}$ and $b_{sca,sample}$ are related to random noise, and hence can be reduced by signal averaging. Since these error components are anyway small relative to the other error components in the model, averaging for 1 min or 1 hour has only a minor effect on the calculated uncertainty in $b_{abs}$.





### 3. Experimental methods

### 3.1 Instrumentation

#### 3.1.1 Instruments for measuring aerosol light absorption and black carbon concentrations

In this Section we detail the experimental methods that we applied to investigate and characterize the ability of the CAPS PMssa to measure atmospheric aerosol absorption coefficients. A total of four different CAPS PMssa monitors were used in this study: one operating at 450 nm, two at 630 nm, and one at 780 nm. The four units are listed in Table 2 along with their relevant specifications.

A multi-angle absorption photometer (MAAP; Thermo Fisher Scientific, Waltham, MA, USA) was used during the Cabauw field campaign (Sect. 3.4.1) to measure absolute aerosol absorption coefficients at a wavelength of 637 nm (Petzold and Schönlinner, 2004). As discussed in the Introduction, the MAAP is a filter-based absorption photometer that incorporates additional measurements of back-scattered light and a two-stream radiative transfer scheme in order to constrain aerosol absorption coefficients more tightly than is possible with simple light attenuation measurements. The MAAP is a well-known and well-characterized instrument for measuring light absorption by atmospheric aerosols. The accuracy of MAAP absorption coefficients was investigated against laboratory reference EMS absorption measurements in the Reno Aerosol Optics Study (RAOS) and the two methods were found to agree within 7% for a range of different black carbon containing aerosols (Petzold et al., 2005). Müller et al. (2011a) demonstrated that the unit-to-unit variability between six different MAAP instruments was less than 5%. These authors also showed that the true operation wavelength of the instrument was 637 nm, not the nominal value of 670 nm. Assuming an absorption Ångström exponent of 1.02, a 5% correction factor should be applied to the firmware output of the MAAP to account for this wavelength difference (Müller et al., 2011a). This correction factor was applied in the present study. During the RAOS campaign (Petzold et al., 2005) MAAP absorption coefficients were observed to have no relationship with aerosol SSA. However, at extremely high SSA values the absorption coefficient measurements from the MAAP can be biased high. To quantitatively compare the MAAP and CAPS PMssa absorption coefficients during the Cabauw field campaign, both coefficients were adjusted to standard temperature (273.15 K) and pressure (1 atm).

A single particle soot photometer (SP2; Droplet Measurement Technologies, Longmont, CO, USA) was used to measure black carbon mass concentrations at high time resolution from a mobile laboratory deployed during the Bologna field campaign (Sect. 3.4.2). The SP2 measures the mass of individual black carbon particles on a single-particle basis using the principle of laser-induced incandescence. The instrument has been described in detail previously (Schwarz et al., 2006; Stephens et al., 2003). Due to its very high sensitivity and responsiveness, its specific purpose in the present study was to provide a high time resolution reference time series of relative absorbing aerosol concentration. Its configuration during the present study is described by Pileci et al. (2020).



### 3.1.2 Particle size classifiers applied for cross calibration and truncation measurements

Two different types of aerosol size classifiers were used to generate monodisperse test aerosols for the purposes of measuring cross calibration constants and scattered light truncation: an aerodynamic aerosol classifier (AAC; Cambustion Ltd, Cambridge, UK) and a differential mobility analyzer (DMA; custom built version of same design as the TSI Model 3081 long-column DMA, TSI Inc. Shoreview, MN, USA). The correct operation and sizing of both types of classifiers was confirmed throughout all the experiments by measuring nebulized PSL particles of different diameters (i.e., by operating the classifiers in scanning mode with downstream concentration measurements performed by a condensation particle counter).

The AAC classifies particles based on their relaxation time under the action of a centrifugal force generated in the annular gap between two rotating coaxial cylinders (Johnson et al., 2018; Tavakoli and Olfert, 2013). The particle relaxation time is related to aerodynamic equivalent diameter in a straight-forward manner. In the context of highly size-dependent optical measurements, the major advantage of such a classification method is that it does not depend on particle electrical charge (unlike the DMA), which means the AAC can produce truly monodisperse distributions of particles (i.e., of finite width but without the presence of additional size distribution modes due to multiply-charged particles). This charge-independent classification approach also enables higher aerosol transmission efficiencies than is possible with the DMA, which improves the signal-to-noise-ratio of any downstream optical measurements. An additional advantage of the AAC relative to the DMA is that it can classify particles over a wider diameter range, including particles with diameters of up to ~5 micrometers. The AAC was operated in the present study with the sheath-to-aerosol flow ratio of around 10:1, which results in geometric standard deviations for the classified aerosols of around 1.14. The set point aerodynamic diameters were converted to volume equivalent diameters using literature values of particle density and assuming the classified particles were spherical.

### 3.1.3 Particle size distributions of ambient aerosol

Measurements of ambient particle size distributions were required during the Cabauw field campaign (Sect. 3.4.1) as inputs for the truncation correction calculations. These measurements were obtained by a scanning mobility particle sizer (modified version of the TSI SMPS 3034; TSI Inc. Shoreview, MN, USA) covering the mobility diameter range from 10 to 470 nm and an aerodynamic particle sizer (TSI APS 3321; TSI Inc.) nominally covering the aerodynamic diameter range from 0.54 to 20 µm.

The hourly-averaged SMPS and APS size distributions were merged to create total aerosol size distributions covering the diameter range from 0.0104 to 10 µm for use in the truncation calculations. This was achieved by first converting the measured diameters of the respective instruments to volume equivalent diameters. The SMPS electrical mobility diameters were simply taken to represent volume equivalent diameter (i.e. shape effects were neglected). The APS aerodynamic diameters were divided by the square root of particle effective density to translate them into volume equivalent diameters. A constant effective



density of 2 g cm⁻³ was assumed. The joined size distributions were then created by linearly interpolating the SMPS and shifted
APS measurements onto a common diameter scale between 0.0104 to 10 μm. The APS measurements of particles with physical
diameters less than 0.6 μm were not used in this joining calculation since they are known to display counting efficiency
problems (Pfeifer et al., 2016).

**3.2 Measurements of scattering cross calibration constants**

Scattering cross calibration constants (Sect. 2.2.2) were measured with the experimental arrangement shown in Fig. S2.
Ammonium sulphate particles or PSL spheres were generated in a Collison type nebulizer and passed through a diffusion drier
filled with silica gel for drying. A filtered bypass line was used after the nebulizer, and the ratio of the flows in this bypass line
and the normal sampling line were adjusted to provide control on the concentration of the nebulized aerosol.

In the default laboratory setup, after drying the particles were passed through a size classifier to produce monodisperse
distributions of particles with modal diameters less than 200 nm (i.e., to produce particles with size parameters less than
approximately 1 that fall within or at least near the Rayleigh light scattering regime, see Sect. 2.2.2). Additionally, to investigate
a simplified procedure for potential application in field campaigns, selected calibrations were also performed by bypassing the
size classifier. In this case only PSL particles with diameters less than 200 nm were produced with the nebulizer to keep the
generated aerosol within or near the Rayleigh light scattering regime. Nevertheless, it is possible that larger PSL aggregates
(doublets or triplets) were also generated by the nebulizer. Such aggregates would be large enough to cause non-Rayleigh light
scattering. In addition, large numbers of non-PSL, smaller particles (most with diameters < ~30 nm with tail extending to 100
nm or larger) are also produced when nebulizing PSL due to the presence of surfactants and other impurities in the PSL and
Milli-Q water solutions. The composition of these particles is generally unknown. The possibility that they contained
substantial absorbing components is unlikely but cannot be ruled out, which would violate the required cross calibration
condition that the calibration aerosol has SSA = 1.

In some of the calibrations a storage volume was placed upstream of the CAPS PMssa unit being calibrated. In these
experiments the volume was first filled with calibration aerosol and the CAPS PMssa was then used to draw the concentration
in the volume down to near zero. This enabled measurement of $\beta_{Rayleigh}$ over a broad range of aerosol loads. In other cases the
calibration aerosol was simply fed directly to the CAPS PMssa unit being calibrated. In all cases we limited the calibration
measurements either during the experiment or later during data processing to $b_{ext}$ values less than 1000 Mm⁻¹ to avoid non-
linearity issues between the scattering and extinction measurements (Sect. 2.2.2).

Two examples of scattering cross calibration measurements are shown in Figs. S3 and S4. Fig. S3 is an example of a calibration
performed with 240 nm PSL particles with the storage volume present to enable measurement across a broad range of aerosol
loadings, while Fig. S4 shows an example where the storage volume was not used such that the measurements only cover a





narrow range of aerosol loading. The 240 nm PSL particles are slightly larger than the particles we typically use for cross calibration but these two examples are shown here to demonstrate the effect of the storage volume. The top panel of these figures show time series of the $b_{ext}$ and $b_{sca,uncalib.}$, and the bottom left panel displays these variables in a scatter plot on a log-
log axis. Onasch et al. (2015) determined $\beta_{Rayleigh}$ as the gradient of a line fit to the scatterplot data. To avoid any potential linear fitting artefacts caused by outlying measurements, we elected to determine $\beta_{Rayleigh}$ as the mean value of the ratio of $b_{sca,uncalib}/b_{ext}$ for $b_{ext}$ values greater than 50 Mm$^{-1}$. This lower limit was chosen to avoid low signal-to-noise ratio measurements affecting the determined $\beta_{Rayleigh}$. The bottom right panel of Figs. S3 and S4 displays histograms of the $b_{sca,uncalib}/b_{ext}$ ratio (with the condition $b_{ext}$ > 50 Mm$^{-1}$). It is seen that the values of the ratio are typically normally distributed, regardless of whether the
measurements covered a broad range of extinction values or not (Fig. S3 vs S4). We take the standard deviation of the measured ratios to represent the precision with which $\beta_{Rayleigh}$ can be determined.

**3.3 Measurements of scattered light truncation as a function of particle diameter**

The general experimental setup that is shown in Fig. S2 was also used to measure scattered light truncation as a function of particle diameter, in order to validate our new truncation calculations (Sect. 2.2.3). Size-resolved truncation measurements can
be performed directly with the CAPS PMssa using size-classified, non-absorbing test aerosols and taking $b_{ext}$ as $b_{sca,true}$, and $b_{sca,Rayleigh}$ as $b_{sca,meas}$ in Eq. (4). Similarly as for the cross calibration constant $\beta_{Rayleigh}$, we applied a threshold condition of $b_{ext}$ > 50 Mm$^{-1}$ when calculating mean ratios of $b_{ext}$ to $b_{sca,Rayleigh}$. For this application, the AAC was always used as the size classifier, since the AAC is able to generate truly monodisperse distributions of particles (i.e., finite width but without additional size modes due to multiply charged particles) and provides a larger upper size limit.


We measured truncation values for three different types of non-absorbing aerosols: PSL, DEHS (Di-Ethyl-Hexyl-Sebacat) and ammonium sulphate. The relevant properties of these aerosols are listed in Table 3. Three aerosol types were used in order to check consistency across different aerosols to provide more robust results. Nebulized and dried PSL and DEHS particles are spherical, while dried ammonium sulphate particles are at least near spherical (e.g. Biskos et al., 2006). Spherical or near
spherical particles were used so that the aerosol phase functions could be calculated precisely with Mie theory. The geometric standard deviations of the monodisperse DEHS and ammonium sulphate aerosols were nominally around 1.14 as determined by the operating conditions of the AAC (Sect. 3.1.2). We used geometric standard deviations of 1.1 in our model calculations for these two aerosol types. The width of PSL size distributions are size-dependent and generally narrower than the transfer function of the AAC as used in these experiments. Therefore, we considered two geometric standard deviations of 1.05 and
1.1 in our model calculations for PSL. Rayleigh normalization factors (i.e., $b_{sca,true}^{Rayleigh}/b_{sca,meas}^{Rayleigh}$, see Eq. 4) were measured at the beginning of each experimental run using particles of the given aerosol type with size parameters less than or close to 1. A number of repeat experiments were performed for some of the aerosol types, as indicated in Table 3.





### 3.4 Field measurements

#### 3.4.1 Cabauw campaign

The Cabauw field campaign was conducted from 11 September to 20 October 2016 at the KNMI (Koninklijk Nederlands Meteorologisch Instituut) Cabauw Experimental Site for Atmospheric Research (the Netherlands; 51° 58' N, 4° 55' E, –0.7 m a.s.l.). The campaign was conducted in the framework of the ACTRIS project (WP11) and occurred simultaneously with the CINDI-2 MAX-DOAS intercomparison campaign (Kreher et al., 2020). The CAPS630b unit was the CAPS PMssa instrument deployed during this campaign to measure absorption coefficients at 630 nm. Absorption coefficients were also measured at

637 nm with a MAAP (Sect. 3.1.1). Particle size distributions were measured with an SMPS and APS (Sect. 3.1.3). All instruments were housed in a laboratory at the base of the KNMI-mast Cabauw behind identical inlets consisting of PM10 sampling hats protruding 4.5 m from the laboratory roof. The inlets contained large diameter nafion driers, which kept RH in the sampling lines below 50%. All the data used in the present study were averaged over one hour periods. This includes the joined SMPS and APS size distributions (Sect. 3.1.3), which used to calculate hourly-resolved truncation correction factors

using the model presented in Appendix A1.

#### 3.4.2 Bologna campaign

The Bologna field campaign was conducted from July 5 to 31, 2017. This campaign was also conducted within the framework of the ACTRIS project. The full campaign consisted of multiple stationary measurement sites that were centered around the city of Bologna in Italy's Po Valley. Additionally, a heavily instrumented mobile measurement van (the MOSQUITA; 

Bukowiecki et al., 2002; Weimer et al., 2009) travelled between the stationary sites to perform spatially-resolved measurements of black carbon concentrations and properties. The results of these mobile measurements are presented by Pileci et al. (2020). In the present study we use only one hour of mobile measurements that were performed from the MOSQUITA while it was travelling on the heavily-trafficked A1 highway between Bologna and Lodi on the morning of July 25, 2017. During this time period absorption coefficients were being measured with the CAPS780, and black carbon concentrations with an SP2 (Sect.

620 3.1.1).

#### 3.4.3 Payerne campaign

The Payerne field campaign was conducted from August 26, 2019 to January 14, 2020 in Payerne, Switzerland and involved the PMssa units CAPS450 and CAPS780. The goal of this campaign was to compare the hygroscopic properties of aerosols measured using remote sensing and in situ techniques. In the present study we only present the results of the CAPS PMssa

cross calibrations that were performed for the campaign: no ambient measurements are shown. In addition to the cross calibrations that were performed at the Payerne field site with the CAPS450 and CAPS780 units, we also present the results of calibrations performed immediately before and after the campaign in the Aerosol Physics Laboratory of the Paul Scherrer Institute. In addition to the two other PMssa units, the CAPS630a was also included in these laboratory calibrations.





## 4. Precision of determination of the cross calibration factor and its stability over time

The cross calibration constants that were measured for the CAPS450, CAPS630a and CAPS780 PMssa units during and around the Payerne field campaign are presented in Fig. 5. These measurements are used to assess the stability of the cross calibration constant (variability over the time series) and the precision with which it can be determined (error bars represent ±1 standard deviation of the ratios of $b_{sca,uncalib.}$ to $b_{ext}$ for each calibration, as visualized in the lower right panels of Figs. S3 and S4). Some of the measurements were performed on size-classified aerosols (solid plot markers), and some were performed without

classification (open plot markers), as discussed in Sect. 3.2

To investigate the effect of size classifying the aerosol, four calibrations were purposely performed back-to-back, with and without an AAC size classifier. The results of these back-to-back calibrations are plotted on their own in Fig. S5. In two cases, the cross-calibration constants determined with and without size classification were similar (CAPS630a with a difference of -

1.9% between calibrations, and CAPS450 run2, with a difference of 2.8%). However, in the other two cases the $\beta_{Rayleigh}$ value determined without size classification was substantially less than the value determined with classification: -9.7% difference for CAPS450 run1, and -6.6% difference for CAPS780. This is likely because of the presence of PSL doublets or triplets, or because the non-PSL particles that are unavoidably generated during the PSL nebulization process either contained absorbing components or were big and abundant enough to cause substantial non-Rayleigh light scattering. In any case, we assume that

the cross calibrations performed with size classification provides the most trustworthy measurement, since it is more certain that all the required conditions for the cross calibration are met.

Despite the potential differences between size classified and non-size classified measurements, some important results are still clearly seen in Fig. 5. Firstly, it is apparent that the variability in $\beta_{Rayleigh}$ over time is instrument-dependent. The different

behaviors observed for these three PMssa units represent the range of performances we have observed for CAPS PMssa monitors in the field. The least stable unit in this context was the CAPS450. In the ten days prior to the beginning of the campaign $\beta_{Rayleigh}$ for CAPS450 was observed to decrease from 0.63 to 0.53. During the campaign itself, $\beta_{Rayleigh}$ ranged from 0.43 to 0.28, showing a general decreasing trend as the campaign progressed. This observed drift corresponds to tens to hundreds of % of uncertainty in $b_{abs}$ (Fig. 4). The CAPS450 instrument diagnostics provided no evidence of instrument

malfunction, change, or contamination during this period. Therefore, this example demonstrates that regular cross calibration validation measurements are necessary to exclude significant drifts. Given that the precise reason for instability in $\beta_{Rayleigh}$ is unknown, we refrain from providing a stability-based uncertainty estimate for this unit in Table 1. However, the four individual AAC-based calibrations performed in the laboratory prior to the beginning of the campaign still provided a chance to investigate the precision-based uncertainty in $\beta_{Rayleigh}$ for CAPS450. The average standard deviation of the ratios of $b_{sca,uncalib.}$

to $b_{ext}$ measured during these four calibrations was 0.009, which is 1.5% of the average $\beta_{Rayleigh}$ of 0.60. Therefore, we conservatively estimate that the $\beta_{Rayleigh}$ can be determined with a precision of 2% for this unit (Table 1).





The averaged value of $\beta_{Rayleigh}$ for CAPS780 was 0.27 over the 14 measurements taken during the 140 days of the campaign (from day 10 to 151 on the cumulative days x-axis). The minimum and maximum values measured during this period were

0.26 and 0.28, respectively. Thus, we conservatively estimate a stability-derived uncertainty value of 8% ($\delta\beta_{Rayleigh}/\beta_{Rayleigh}$) for this unit in Table 1, while noting that this estimate is derived from calibration measurements without a size classifier, which may have contributed to the observed variability. The 8% uncertainty range is represented by the green-shaded uncertainty band in Fig. 5. The precision-based uncertainty estimate in $\beta_{Rayleigh}$ for this unit is determined from the five AAC-based cross calibrations performed in the laboratory before and after the campaign (i.e., the average size of the green

error bars). From these measurements we calculate a precision-derived uncertainty of 6%. The CAPS630a was not operated during the Payerne field campaign period. However, laboratory measurements before and after the campaign indicated that $\beta_{Rayleigh}$ for this unit can be determined with a very high precision of 2% and is stable to within 2% over time. We believe that this unit represents an example of the best case performance for cross calibration precision and stability that is possible with the CAPS PMssa.


In addition to continual drifts in $\beta_{Rayleigh}$ over time, it is also interesting to note how $\beta_{Rayleigh}$ can change following known events such as contamination of the PMssa optical cavity. Fig. S6 displays CAPS PMssa measured $b_{ext}$ (left panel) and $b_{sca}$ (right panel) at 450 nm against independent measurements of these quantities (CAPS PMex for $b_{ext}$, nephelometer for $b_{sca}$) during a field campaign at the rural background site of Melpitz, Germany. Throughout the duration of these measurements the optical

cavity of the 450nm PMssa unit became contaminated, with average baseline optical loss varying from 758 to 1248 Mm⁻¹. This caused an increase in the bias of the $b_{ext}$ measurement relative to the corresponding PMex measurement from 5 to 17%. Over the same period, the bias of the $b_{sca}$ measurement with respect to the corresponding nephelometer measurement was unchanged. This suggests that contamination of the optical cavity of the PMssa unit created a bias in the instrument's $b_{ext}$ measurement by CAPS. We hypothesized that the contamination caused an increase in the total optical loss of the CAPS,

resulting in the CAPS measurement entering the nonlinear regime normally described by $b_{ext} > 1000$ Mm⁻¹. However, inspection of the data did not support this hypothesis. We therefore recommend that the CAPS PMssa baseline should be monitored continuously throughout measurement campaigns for signs of mirror contamination, and that contaminated mirrors are cleaned promptly.

### 5. Laboratory truncation measurements and comparison against model calculations

The results of the laboratory truncation measurements as a function of AAC-selected particle diameter are shown in Fig. 6 for both PSL and DEHS test aerosols. We refer to these curves as 'truncation curves'. Truncation curves are a useful way to validate truncation calculations since the particle size is a key determinant of the aerosol scattering phase function and consequently γ for spherical particles of known constituent material. Following earlier studies (Liu et al., 2018; Onasch et al.,



2015), we display measured truncation values as the inverse of *γ* as defined by Eq. (4). Measurements are presented at three

different wavelengths (corresponding to the three figure columns) as measured by the CAPS450, CAPS630a and CAPS780 PMssa units. The equivalent measurements for ammonium sulphate are shown in Fig. S7. These results are not included in Fig. 6 since we suspect that the ammonium sulphate particles were slightly non-spherical after nebulization and drying, which makes them less useful for comparison with Mie-theory based model curves, as is done below. Nevertheless, it is seen that the ammonium sulphate measurements are qualitatively consistent with the PSL and DEHS results over many repeated

experiments. The PSL measurements at 450 and 630 nm presented by Onasch et al. (2015) are also included in Fig. 6. They indicate less truncation than the corresponding measurements from the present study. The reasons for these discrepancies are not entirely clear but may be related to the fact that the Onasch et al. (2015) measurements were obtained after size classification by DMA, although the authors found no substantial evidence of additional size distributions peaks due to multiply charged particles (and such particles would anyway cause greater truncation, not less).


A variety of modelled truncation curves are also displayed in each panel of Fig. 6 for comparison with the measurements. Broadly, these can be classified into calculations that include the process of scattered light reflection from the inner surface of the glass sampling tube, and those that do not. One uncertain parameter is the extra path length outside the integrating sphere that contributes to scattered light collection (the *l* parameter), which was set to 1 cm in the original model calculations by

Onasch et al. (2015) without considering glass tube reflection (dashed grey lines in Panels a and b). The corresponding truncation curves calculated with the new model presented in Appendix A1 and with the process of glass tube reflection switched off (solid light grey lines) agree well with the original model. Calculations made with the new model with the process of glass tube reflection turned on and *l* set to 1 cm are shown as the solid colored lines. The shaded envelopes around these curves demonstrate the sensitivity of modelled truncation to variation of *l* between its lower and upper geometrical boundaries

(0 cm ≤ *l* ≤ 4.7 cm). Finally, the dashed colored curves in each panels are truncation curves calculated with the RTE-based model presented by Liu et al. (2018). These curves include the process of glass tube reflection and assume *l* = 1 cm.

Measured and modelled truncation curves all display the same general features. Truncation values are relatively flat up to a volume equivalent diameter of around 200 nm (or ~150 nm at the 450 nm wavelength). This corresponds to the approximate

limit of the Rayleigh light scattering regime. At larger diameters, particles begin scattering relatively more light in near-forward directions, where it escapes from the CAPS PMssa integrating sphere. As a result of this loss of scattered light, the truncation curve begins decreasing with increasing particle diameters in a complicated but well-known manner due to the variation of the scattering phase function with particle diameter. Specifically, at diameters >~ 1 μm, peaks in the truncation curves occur due to Mie resonances. The peaks are slightly broadened by the small polydispersity of the AAC-selected size distributions. This

can be seen when comparing the Onasch et al. (2015) modelled curves, which were calculated assuming a perfectly monodisperse distribution of particles, and the modelled curves from the present study, which consider the finite width of the experimental size distributions (in the case of PSL two geometric standard deviations of 1.05 and 1.1 are modeled). The





remarkable fact that the Mie resonances are discernable in the measured PSL and DEHS truncation curves, even if perfect
quantitative agreement isn't obtained with the models, provides high confidence in the AAC-CAPS PMssa setup for measuring
scattered light truncation. The Mie resonances are not as apparent in the ammonium sulphate measurements, which we suspect
is likely due to particle non-sphericity as mentioned above.

In general, good agreement is obtained between the new model calculations including the process of glass tube reflection and
the RTE model calculations. This is encouraging given these models are based on fundamentally different approaches for
calculating truncation. Both of these modelled curves predict generally greater truncation than the calculations that neglect the
process of glass tube reflection, as expected from theoretical considerations (Sect. 2.2.3). The measurements are in better
agreement with the modeled curves that include glass tube reflection, demonstrating that this process must be considered in
the calculations. This is a robust result that is consistent across all three aerosol types for particle diameters up to 5 μm and all
three optical wavelengths that were investigated.

Considering the calculations made with the new model presented in Appendix 1, the best agreement with the measured data
appears to be obtained for an $l$ value of 1 cm. However, there is enough scatter in the measurements to argue that any $l$ value
is plausible within the geometric limits of this parameter (from 0 to 4.7 cm). Although varying $l$ over this range captures the
measurements well, we stress that this does not imply that variable $l$ is the physical reason for the measurement imprecision.
For one, it can be argued that the lower limit $l$ value of 0 cm is unrealistic since it is expected that particles at the boundary of
the sphere will certainly scatter light into the sphere. Similarly, our model uses an idealized geometry of the interior of the
CAPS PMssa cell. Nevertheless, varying $l$ from 0 to 4.7 cm produces differences in calculated truncation that are similar to
the differences observed between repeat measurements, as well as to the differences between calculations made with the two
models that include the process of glass tube reflection (i.e., the model presented in Appendix 1 and the RTE model). Therefore,
similarly to Onasch et al. (2015), we use $l$ as a convenient tuning parameter to produce an uncertainty envelope that captures
the range of measured truncation curves reasonably well.

### 6. Examples from the field: measurements of atmospheric aerosol absorption coefficients with the CAPS PMssa

### 6.1 Bologna example: instrument sensitivity and rapid response time

Two key features of the CAPS PMssa as a flow-through, continuously-measuring optical instrument are its sensitivity and
rapid response time. Onasch et al. (2015) demonstrate the instrument is able to respond to changes in $b_{ext}$ and $b_{sca}$ of less than
1 Mm$^{-1}$ on timescales of only ~1 second. These specifications suggest that EMS-derived $b_{abs}$ values measured by CAPS PMssa
will display similar responsiveness and sensitivity. If so, these specifications would represent a major improvement over the
equivalent specifications for $b_{abs}$ values measured by filter-based absorption photometers, which are based on the slower
process of accumulation and detection of aerosol samples on a filter.






To investigate these features under real-world conditions, Fig. 7 presents co-located measurements of aerosol absorption coefficients obtained with the CAPS780 instrument and rBC mass concentration measurements obtained with an SP2. The measurements were obtained while travelling along a busy highway road in a mobile laboratory near Bologna, Italy, where black carbon was shown to be the dominant source of absorbing aerosol. Indeed, the observed correlation between the

independent measurements of black carbon and aerosol absorption is remarkably good, especially given the short averaging time of 5 seconds. In particular, the sharp peaks that are observed in both time series are found to align with each other extremely well. These peaks correspond to black carbon emissions from passing vehicles on the highway. Although this is only a one-hour sample of data, this example demonstrates the EMS-derived $b_{abs}$ values can be measured at very high time resolution with the CAPS PMssa, comparable to what can be achieved with the single-particle level measurements of rBC

mass from an SP2.

Although Fig. 7 demonstrates the responsiveness of the CAPS PMssa, it must be stressed that the absolute values of the plotted $b_{abs}$ measurements are still uncertain and should not be used quantitatively. Specifically, the plotted quantity is $b_{abs,Rayleigh}$ to indicate that the underlying $b_{sca}$ measurements have not been corrected for the truncation of non-Ralyeigh scattered light. To

do so accurately would require co-located, equally high time resolution measurements of either scattering phase functions or information that could be used to calculate phase functions (e.g. size distributions, fractal BC properties) for the freshly emitted BC-containing emissions plumes. Nevertheless, the ability to measure relative $b_{abs}$ values at high time resolution with the CAPS PMssa creates possibilities for new types of experiments that were not previously feasible with traditional absorption photometers.


### 6.2 Cabauw example: comparison of CAPS PMssa absorption measurements against independent measurements with a MAAP

In this section we use an example dataset from the Cabauw campaign to perform a full quantitative assessment of the ability of the CAPS PMssa to measure absolute aerosol absorption coefficients, including a detailed characterization of the scattered

light truncation effects, as well as the uncertainties in the other underlying measurements. The CAPS630b was the PMssa unit operated during this campaign. This unit ran autonomously, continuously, and stably over the one month of operation, which we have found to be typical for CAPS PMssa units operated at stationary field sites.

### 6.2.1 CAPS PMssa baseline characteristics

The scattering and extinction baseline measurements over the campaign are displayed in Fig. S8. These measurements were

performed for 1 min every 10 mins using the auto-baselining feature of the instrument. The average standard deviations of the extinction and scattering baseline measurements over all 1 min baseline periods were 0.35 and 0.66 Mm$^{-1}$, respectively. We





take these values to represent the precision-based uncertainty estimates in $b_{ext,baseline}$ and $b_{sca,baseline}$ in Table 1. Both baseline time series indicate that the optical cavity was generally clean and suffered no major contamination events during the campaign. Fig. S8 also contains time series of the baseline drift in each channel, which were calculated from the differences between two

successive baseline measurements (i.e., over a period of 10 mins, which means the calculated metric does not include possible variations over shorter time scales). On average, the extinction baseline drifted by 0.026 Mm$^{-1}$ min$^{-1}$, and the scattering baseline by 0.013 Mm$^{-1}$ min$^{-1}$. Individual values of up to 0.3 and 0.09 Mm$^{-1}$ min$^{-1}$ were observed in the extinction and scattering channels, respectively. To minimize the impacts of these drifts, both the extinction and scattering measurements were reprocessed using the method of linear interpolation between successive baseline measurements (Pfeifer et al., 2020). Fig. S9

indicates that this reprocessing had a noticeable effect at a 1-second time resolution for $b_{ext}$ values less than ~20 Mm$^{-1}$ and $b_{sca}$ values less than ~5 Mm$^{-1}$. However, these effects are averaged out when considering hourly-averaged data.

### 6.2.2 Uncertainties in the CAPS PMssa $b_{sca}$ correction factors $\beta$ and $\gamma$

The cross calibration constant for the CAPS630b could be measured with high precision (~2%) and appeared to be very stable based on measurements at the beginning (0.82; Fig. S10) and at the end of the campaign (0.80; Fig. S11), which differed by

only 2.5%. This is comparable with the optimum performance we have observed for CAPS PMssa units with regards to cross calibration (see Sect. 4).

In the absence of direct scattering phase function measurements, hourly time resolved truncation correction factors were calculated with the Mie-theory based model presented in Appendix A1. Joined size distributions measured by SMPS and APS

were input into the model. To estimate the uncertainty in $\gamma$ a sensitivity analysis was performed with respect to the model's input parameters. This analysis is presented in the Supplementary Information as Sect. S1 and visualized in Fig. S12. Specifically, we investigated the sensitivity of $\gamma$ to the ambient aerosol refractive index (real parts between 1.50 to 1.59, and imaginary parts between 0.00 to 0.01, based on the summary of measurements presented by Espinosa et al. (2019), the $l$ parameter (varied from 0 to 4.7 cm based on the results presented in Sect. 5), and the accuracy of the coarse mode size

distribution measurements between diameters of 2.5 and 10 μm. The truncation correction was found to be most sensitive to the $l$ parameter (~4% across the range of tested parameters), weakly sensitive to the real part of the refractive index and size distribution information between 2.5 and 10 μm, and barely sensitive at all to complex part of the refractive index (not shown). Overall, we estimate an uncertainty in $\gamma$ of 6% for the Cabauw campaign based on this analysis (difference between the minimum and maximum average values of all the simulated distributions shown in Fig. S12). However, it must be stressed

that this estimate is limited by our Mie-theory based calculations, which do not include potential effects due to morphologically-complex particles such as freshly-emitted fractal BC aggregates. That is, the estimate only covers the parametric uncertainty in our Mie-theory based calculations.





### 6.2.3 Truncation effects for fine-mode dominated and coarse-mode containing samples

The distributions of the time-resolved truncation correction factors calculated over the Cabauw campaign are clearly bi-modal

(Fig. S12). This indicates that there were two limiting types of aerosols measured during the campaign: i) fine-mode dominated aerosol that only required a minor truncation correction; and ii) aerosol with substantial coarse mode fraction that required a larger truncation correction. For further investigation, we extracted two subsets of data representing the fine-mode dominated and coarse-mode containing samples. This was done by selecting those aerosols whose coarse-mode number fractions (defined as $N_{D_p>1\mu m}/N_{total}$ calculated from size distributions) are in the lower and upper quartiles of all the data, respectively. The

median normalized size distributions for these two categories are plotted in Fig. S13. The figure clearly shows a coarse mode of particles with diameters between ~0.6 and 5 µm that was present in the coarse-mode containing samples but not the fine-mode-dominated aerosols. Despite the small number fractions, these coarse mode particles can make substantial contributions to the scattering coefficients and they produce a greater fraction of forward and backward scattered light, thereby affecting truncation disproportionately.


The distributions of the time-resolved $\gamma$ values for the fine-mode dominated and coarse-mode containing groups of samples are displayed in Fig. S14 (for the same ranges of model inputs that were examined for the full Cabauw dataset in Fig. S12). As expected, the required truncation correction values are substantially smaller for the fine-mode dominated group than the coarse-mode containing samples. This follows from the normalized ensemble scattering phase functions for each group, which

are displayed in the right panel of Fig. S13. These functions were calculated from the median size distributions with Mie theory and an assumed refractive index of $1.59 + 0.01i$ (functions like these are required as inputs for the $\gamma$ calculation, as shown in Fig. 3). It is seen that the phase function for the fine-mode dominated category is less focused in the near-forward (scattering angles < 25°) and backward (scattering angles > 125°) scattering directions than that of the coarse-mode containing category, which is why these samples are associated with lower $\gamma$ values.


Fig. S13 also compares the Mie calculated scattering phase functions with measurements obtained by Espinosa et al. (2018) from a broad range of aircraft flights conducted throughout the USA. Three categories of measurements are shown: two categories of coarse-mode containing aerosols (coarse categories 1 and 2) and one category of 'fine' aerosols (which is comprised of measurements for aerosols classified as 'urban', 'biomass burning' and 'biogenic', all of which were observed

to have very similar scattering phase functions). The calculated scattering phase functions agree reasonably well with the corresponding measurements at near-forward scattering angles (comparing the fine-mode dominated and 'fine' categories; and the coarse-mode containing and the two coarse categories). However, at near-backward scattering angles the Mie calculations predict proportionally more light scattering than is observed in the measurements. If it is assumed that the measurements by Espinosa et al. (2018) are reasonably representative of the average scattering phase functions of these aerosol types also in

European continental air masses, this comparison suggests that the Mie calculations would tend to slightly overestimate the



truncation correction factors displayed in Figs. S12 and S14, and that the degree of overestimation would be greater for the coarse-mode containing group than the fine-mode dominated samples. Considering only the parametric uncertainty in the calculated $\gamma$ values (i.e., that uncertainty related to the Mie truncation model inputs), we estimate values of 4% and 9% for the fine-mode dominated and coarse-mode containing groups, respectively, based on the minimum and maximum mean values

for each of the groups shown in Fig. S14.

### 6.2.4 Comparison of CAPS PMssa and MAAP $b_{abs}$ measurements

Considering all of the underlying CAPS PMssa uncertainties for the Cabauw dataset that are summarized in Table 1, it is clear that the largest individual source of uncertainty is related to the truncation correction that must be applied to $b_{sca}$. To investigate the effect of this uncertainty on the ultimate derived $b_{abs}$, Fig. 8 compares these measurements against independent $b_{abs}$

measurements obtained with a MAAP under three different truncation correction scenarios (corresponding to the three rows of the figure). It is worth recalling that the coefficients from both instruments were adjusted to standard temperature (273.15 K) and pressure (1 atm) for this quantitative comparison. The plot is further split into three columns, following the data grouping done in the previous subsection, so that the results for the fine-mode dominated samples, the coarse-mode containing samples, and the full dataset taken as a whole can be inspected separately. Each subplot contains an uncertainty envelope that

represents the 95[th] percentile of hourly-resolved theoretical uncertainty values calculated with the error model and inputs presented in Sect. 2.3 and Table 1. Uncertainties in $\gamma$ of 4, 9, and 6% were used for the fine-mode dominated samples, coarse-mode containing sample and full dataset, respectively.

The first row of this figure (Figs. 8a, b, and c) displays the CAPS PMssa $b_{abs}$ measurements that were processed with time-

resolved truncation correction factors calculated with the Mie-theory based model (Appendix A1) with $m$ set to $1.59 + 0.01i$ and $l$ to 4.7 cm. This processing results in generally good agreement between the CAPS PMssa and MAAP $b_{abs}$ measurements. Strong correlation is seen between the two independent measurements for all three subsets of the data (Figs. 8a, b, and c). However, on average, the absolute values of the two measurements are systematically offset by about 20%. In particular, the mean ratios of CAPS $b_{abs}$ to MAAP $b_{abs}$ (indicated by the solid orange line in each sub-figure) varies from 0.81 to 0.84. The

precise reasons for these systematic offsets are not clear. One possibility is the geometry correction factor applied to the CAPS630b PMssa data (0.7) was inaccurate. However, we consider it unlikely that this factor alone could fully explain the observed discrepancy (it would need to be lower by ~20% – since $b_{abs}$ is directly proportional to $1/\alpha$, Eq. (7) – and this is beyond the range of $\alpha$ values that have so far been measured for PMssa units, see Sect. 2.2.1). Importantly, the bias between the CAPS PMssa and MAAP measurements displays no, or only minor, dependence on SSA (which would be detectable as

colour trends across the scatter plots). This indicates that it is likely that the CAPS PMssa $b_{sca}$ measurements have not been over-corrected for truncation, since such over-correction would affect the high SSA samples more than the low SSA samples. For these reasons, we believe that Figs. 8a, b, and c represent an example of reasonably well estimated truncation correction.


We note that this result was achieved with an $l$ value of 4.7 cm, which is within the plausible range for this model input parameter based on the results presented in Sect. 5.


The second row of the figure (Figs. 8d, e, and f) displays measurements corrected with time-resolved $\gamma$ values calculated with the Mie model with $l = 1$ cm, which represents the best guess for this parameter based on the laboratory truncation curve measurements presented in Sect. 5. Under this truncation correction scenario, a sizeable number of the hourly-averaged coarse-mode containing CAPS $b_{abs}$ measurements shown in Fig. 8e are negative because $b_{sca} > b_{ext}$, which is physically not possible.

In addition, the bias between the CAPS PMssa and MAAP measurements displays an SSA dependence, most clearly seen in Figs. 8e and f. Together, these pieces of evidence indicate that the CAPS $b_{sca}$ measurements have been slightly overestimated, leading to absolute $b_{abs}$ values that are biased low. Despite the poor agreement between the absolute CAPS PMssa and MAAP values for this truncation scenario, it is noteworthy that the two measurements still correlate well with each other.

The third and final row of this figure (Figs. 8g, h, and i) displays measurements that have been corrected for Rayleigh light scattering only. This means a constant $\gamma$ value of 1 was applied to the $b_{sca}$ measurements. As shown in Fig. 2, this is the same as simply taking the $b_{sca}$ values output by the instrument firmware (i.e., $b_{sca,Rayleigh}$), assuming that the firmware contained the correct $\alpha$ and $\beta$ values. Here it is clear that the truncation correction has now been underestimated, leading to $b_{abs}$ values that are greater than the corresponding y-axis values in the first row of the figure. Again, this is most clearly seen for the

coarse-mode containing group of samples (Fig. 8h). On average, the $b_{abs}$ values for these samples are 1.94 times higher than the corresponding MAAP measurements.

To further examine the sensitivity of CAPS PMssa $b_{abs}$ measurements to the scattering truncation effect, five additional truncation correction scenarios are displayed in the five rows of Fig. S15. These results generally support the results displayed

in Fig. 8. Additionally, the following conclusions can be drawn: (i) for a given $l$ value (4.7 cm in this case), variation of the ambient aerosol refractive index between $1.5 + 0.01i$ and $1.59 + 0.01i$ has only a minor effect on CAPS PMssa measured $b_{abs}$, which can be seen by comparing Figs. S15a-f and Figs. 8a-c, (ii) setting $l$ to 0 cm in the model (the lower limit for $l$ that we derived from Fig. 6) results in substantially overestimated truncation and $b_{sca}$, which in turn leads to a substantial fraction (35.3%) of negative $b_{abs}$ values for the coarse-mode containing group (Fig. S15h), and (iii) using constant $\gamma$ values calculated

from campaign-averaged size distributions (the averaged joined SMPS and APS particle size distribution, Figs. S15j-l; and the averaged SMPS distribution only Figs. S15m-o) generally leads to poorer results for the coarse-mode containing samples compared to the corresponding results obtained with time-resolved truncation correction, but the results for the fine-mode dominated samples are similar. This latter result is pertinent to field studies where time-resolved scattering phase function information is not readily available (either directly or indirectly, e.g. in the form of measured size distributions), such that a

user might be forced to use a constant truncation correction factor.



### 6.2.5 Summary of the Cabauw results

The Cabauw example demonstrates that the biggest hurdle that must be overcome when measuring atmospheric aerosol absorption coefficients with the CAPS PMssa is accurate accounting of the scattered light truncation effect. Unfortunately, there are many potential sources of errors in $\gamma$, as discussed in a general sense in Sect. 2.3. Even if the scattering phase functions
for the atmospheric aerosols being measured were known with high accuracy, $\gamma$ would still carry substantial uncertainty related to the instrument geometry that must be considered in the truncation calculation, which we choose to represent through the glass sample tube extension length $l$. This uncertainty can lead to a broad range of different final $b_{abs}$ outcomes, as we have shown by examining the sensitivity of $b_{abs}$ to variation in the $l$ parameter.

Although the uncertainty in $\gamma$ cannot be totally avoided, its effect on $b_{abs}$ can be substantially mitigated by restricting datasets to only those aerosol samples that do not display strongly forward-focused light scattering. For the Cabauw example, we successfully achieved this by separately analysing the fine-mode dominated samples. The results for this group of samples were generally very consistent over the range of truncation correction scenarios we investigated. Considering the cases shown in Figs. 8 and S15, the correlation coefficients between CAPS PMssa and MAAP $b_{abs}$ varied between 0.91 and 0.96, while the
average ratios of the two measurements varied from 0.64 to 0.99 (for a sample size of 150). Although the precise reasons for the systematic offset between the CAPS PMssa and MAAP measurements are still unclear, the consistency of the results against the changes in $\gamma$ is highly encouraging. This suggests that despite the remaining truncation uncertainties, the CAPS PMssa can still provide a reliable $b_{abs}$ measurement for fine-mode dominated atmospheric aerosols.

In contrast, the equivalent results for the coarse-mode containing samples were more problematic. For this group and for the truncation correction scenarios shown in Figs. 8 and S15, the correlation coefficients between CAPS PMssa and MAAP $b_{abs}$ varied between 0.88 and 0.95, while the average ratios of the two measurements varied widely from 0.03 and 1.94 (again for a sample size of 150). The truncation problems for coarse-mode containing samples are two-fold. Firstly, the scattering phase functions for such samples are highly asymmetric, with enhanced forward- and backward-scattering, which means the
corresponding $\gamma$ values are large and more sensitive to small changes in particle size, shape, and/or composition (as demonstrated with respect to particle size, for example, by the steepness of the truncation curves displayed in Fig. 6 at particle diameters greater than ~1 µm). This sensitivity is likely responsible for the large variability in the mean ratios of CAPS PMssa to MAAP $b_{abs}$ that were obtained for the coarse-mode containing samples across the tested truncation scenarios.

Adding to this is a second problem, which is that the Mie-theory-predicted phase functions are likely to be inaccurate for super-micrometer particles of complex morphology (e.g. mineral dust aerosols; Curtis et al., 2008). This problem could potentially be overcome by performing truncation calculations with scattering phase functions that have been measured directly, or calculated with consideration of complex particle morphologies (e.g. using more sophisticated optical models, or potentially





with scattering phase functions parameterized according to the asymmetry parameter, as inspired by the truncation
relationships presented by Liu et al., 2018). However, further work is required to determine how much such truncation
correction methods could improve the reliability of CAPS PMssa measurements of aerosols with substantial coarse-mode
number fractions.

**7. Conclusions and recommendations for future studies that use the CAPS PMssa to measure absorption coefficients**

We have developed a detailed error model for the CAPS PMssa (Sect. 2.3) and used this as a framework for assessing the
ability of the instrument to measure aerosol absorption coefficients via the EMS method In combination with empirical data,
this error analysis underlines the importance of minimizing errors in $b_{ext}$ and $b_{sca}$. Two key sources of error were identified as
requiring further investigation: uncertainties in the instrument cross calibration constant and those related to the truncation
correction. Properly accounting for scattered light truncation is the more difficult problem. Our laboratory measurements
demonstrate that the process of glass tube reflection (Liu et al., 2018) must be considered in the truncation calculation (Sect.
5). This process was neglected in earlier truncation models (Onasch et al., 2015). However, uncertainty still remains regarding
the length of the optical cavity that should be considered in the calculation. Furthermore, if co-located scattering phase
functions cannot be measured directly for input to the calculation, one must carefully consider the large range of potential
errors that can arise in calculated scattering phase functions. The uncertainties in the cross calibration constant are less
problematic. The cross calibration constants can be measured with high precision but regular measurements are required to
identify potential drifts. The required frequency of regular cross calibrations varies between instruments (Sect. 4).

We presented two example field datasets to illustrate the potential and limitations of using the CAPS PMssa to measure
atmospheric aerosol absorption. The first example from Bologna demonstrates that the CAPS PMssa can be used to provide
much higher time resolution measurements of relative absorption coefficients than is possible with filter-based absorption
photometers. The second example from Cabauw confirms that a proper truncation correction is the biggest hurdle to overcome
to accurately measure $b_{abs}$ for atmospheric aerosols with the CAPS PMssa. Nevertheless, we demonstrated that under certain
conditions – in this case when fine, submicrometer aerosols dominated the particle size distributions – the CAPS PMssa
provides consistent EMS measurements over a range of different truncation scenarios, even for SSA values greater than 0.95.

Based on the lessons learned in the present study, we recommend that the following steps be taken in future studies that use
the CAPS PMssa to measure aerosol absorption with the EMS method. Although our focus has been on atmospheric
measurements, these recommendations are also applicable to other types of experiments, such as emissions testing and other
laboratory experiments. Furthermore, although our focus has been heavily focused on $b_{abs}$ where the consequences of errors
are the most severe, these recommendations also apply to CAPS PMssa measurements of $b_{ext}$, $b_{sca}$, and SSA.






- Accurate knowledge of the geometry correction factor, $\alpha$ for a given unit is essential for performing accurate absolute measurements. Due to the manner in which the CAPS PMssa is cross calibrated, both $b_{sca}$ and $b_{abs}$ are directly proportional to $\alpha$, and relative errors in $\alpha$ propagate linearly to errors in $b_{sca}$ and $b_{abs}$. Although its seems that the $\alpha$ for a given unit is relatively stable over time, an obvious future improvement to the instrument would be to include a diagnostic for monitoring $\alpha$ during instrument operation (e.g. by measuring and recording the instrument purge flows).

- The periodic scattering and extinction baseline measurements that are performed by the CAPS PMssa over the course of a measurement campaign should always be inspected (e.g. Fig. S8), to ensure reliable instrument operation and to check for the occurrence of potential contamination events. Contamination can drastically reduce the instrument sensitivity and also alter a unit's cross calibration constant (Fig. S6).

- Following Pfeifer et al. (2020), we recommend that both the scattering and extinction coefficients should be reprocessed with linear (or cubic spline) interpolation between successive baseline periods, rather than relying on the default firmware method of step-function interpolation. The Cabauw results at an optical wavelength of 630 nm indicated that such reprocessing was not important at the hourly-time resolution level, since the impact of baseline drifts on shorter time scales cancelled each other out. Nevertheless, the reprocessing should always be done as a precaution against rapidly changing carrier gas compositions, particularly for lower wavelength units (e.g. 450 and 530 nm) operating in urban settings or other situations where high $NO_x$ concentrations might be encountered (Pfeifer et al., 2020).

- The scattering cross calibration constants for some PMssa units can be measured with sufficiently high precision (~2%) and are stable over time (~2%) for accurate EMS measurements. However, for other units the performance can be poorer, especially with respect to stability. Regular cross calibrations should be performed for each individual unit to determine its behavior in this regard. This information should be used to inform experimental designs (e.g. required frequency of cross calibrations for individual PMssa units).

- When calculating scattered light truncation, a model should be used that includes the process of reflection from the inner surface of the glass sampling tube. Uncertainty still remains regarding the choice of the $l$ parameter – the extra path length outside the integrating sphere to be considered in the truncation calculation. In the absence of a suitable independent reference, we recommend setting $l$ to 1 cm, which was the value that resulted in the best agreement between the measured and modeled truncation curves displayed in Fig. 6. However, an uncertainty band formed by varying $l$ between of 0 and 4.7 cm should be considered. If an independent reference point is available for a particular experiment, these measurements can be used to assess the most appropriate $l$ value. The Cabauw dataset provided an example of how this can be done using co-located MAAP absorption measurements (Sect. 6.2.4).

- Uncertainty in the truncation correction also results from the treatment of the scattering phase function.
  - o In the ideal case, the ensemble scattering phase functions should be obtained directly with co-located polar nephelometer measurements. Future studies should be performed to determine if such co-located measurements will enable the CAPS PMssa to reliably measure absorption coefficients even for aerosols



1020        containing high fractions of particles with highly asymmetric scattering phase functions (e.g. super-micrometer particles generally, fractal BC, dust).

     o   If phase functions have to be calculated or assumed, then the aerosol sample to be measured should be conditioned to ensure that its scattering phase function is not too highly forward focused. For the Cabauw example, this was achieved in the post-processing stage by separately analyzing the fine-mode dominated

1025        samples. It could also be achieved at the measurement stage. For example, if the light-absorbing particles of interest reside primarily in the fine mode, a PM1 selective inlet could be placed upstream of a CAPS PMssa unit to ensure that it only measures sub-micrometer particles. These approaches will not eliminate the uncertainties in the truncation calculation, but they can mitigate the influence of those uncertainties on the precision of the derived $b_{abs}$ values.


Regarding the final point, the influence of fractal BC aggregate particles deserves special mention. These particles constitute one of the key types of absorbing aerosols, especially in field or test-bench measurements of fresh emissions from combustion sources. Fractal aggregates scatter more light into near-forward directions relative to equivalently sized spherical particles, in a manner that cannot be predicted by Mie theory (Liu and Mishchenko, 2007). Even when more advanced scattering

calculations are performed, the morphology (e.g. primary particle sphere size and fractal dimension) of fractal aggregates is often difficult to constrain. These effects create the potential for large and systematic errors in calculated $\gamma$ values, which sets up a trade-off when measuring aerosols with high proportions of fractal BC aggregates with the CAPS PMssa. As the fraction of absorbing fractal BC increases, SSA decreases, which decreases subtractive error amplification (Fig. 4). However, at the same time, errors in $\gamma$ likely increase due to the shift in the scattering phase function towards forward directions, which would

at least partially offset the reduction in error due to the lower SSA. Therefore, it should not be assumed that errors in EMS-derived $b_{abs}$ will always be necessarily lower for aerosols of low SSA. Knowledge of the magnitude and even signs of the potential errors in $\gamma$ due to the presence of absorbing fractal aggregates is currently very limited and further studies are required to investigate this issue in more detail.

**8. Appendix A1: New theoretical model for calculating scattered light truncation in the CAPS PMssa monitor including**
**the process of glass tube reflection**

**8.1 General formulation for describing aerosol scattering coefficients measured by integrating nephelometers**

An integrating nephelometer measures the integrated particulate scattering coefficient of an aerosol sample $b_{sca}$ (also typically denoted as $\sigma_{sp}$) at a given wavelength $\lambda$. An ideal integrating nephelometer would be sensitive to light scattered in all possible directions. Formally, if $S_p(\lambda, \Omega)$ (m$^{-1}$ sr$^{-1}$) was a function describing the distribution of scattered light of wavelength $\lambda$ as a

function of solid angle $\Omega$ for some aerosol, an ideal integrating nephelometer would collect light scattered over all $4\pi$ stearadians with equal sensitivity,





$$b_{sca}(\lambda) = \int_0^{4\pi} S_p(\lambda, \Omega) d\Omega = \int_0^{2\pi} \int_0^{\pi} S_p(\lambda, \theta, \phi) \sin(\theta) \, d\theta d\phi \quad , \tag{A1}$$

where $\theta$ and $\varphi$ represent the polar and azimuthal scattering angles, respectively, in a polar coordinate system. If the scattering
process is rotationally symmetric with respect to the azimuthal coordinate – which is true for the case considered here of
spherically homogeneous particles illuminated by unpolarized light (Mishchenko et al., 2002) – the integral over $\varphi$ equals $2\pi$
and

$$b_{sca}(\lambda) = 2\pi \int_0^{\pi} S_p(\lambda, \theta) \sin(\theta) \, d\theta \, . \tag{A2}$$

In practice, real integrating nephelometers are unable to collect some fraction of near-forward and near-backward scattered
light due to physical design limitations. This issue is known as scattered light truncation. If truncation is not accounted for then
particulate scattering coefficients measured with an integrating nephelometer will be systematically underestimated. In the
Rayleigh regime (particle diameter $D_p \ll$ the wavelength of light $\lambda$), truncation is independent of $D_p$ and particle shape. In the
Mie regime ($D_p \sim \lambda$), truncation is a complicated function of $D_p$ and particle shape, with larger particles tending to produce
larger truncation as the fraction of light scattered in forward directions with small $\theta$ increases.

**8.2 Calculating scattered light truncation in the CAPS PMssa**

The CAPS PMssa measures particulate scattering coefficients at a single wavelength $\lambda$ with an integrating nephelometer of the
reciprocal, integrating sphere design (Heintzenberg and Charlson, 1996). The integrating sphere has a nominal diameter $L =$
10 cm. Aerosol particles with number size distribution $dN/d\log D_p$ travel through the sphere along a central axis in a horizontal,
cylindrical glass tube of nominal diameter $d = 1$ cm. Light scattered from the aerosol ensemble is detected with a
photomultiplier tube (PMT) placed at one point on the integrating sphere.

To correct for the scattered light truncation effect in the CAPS PMssa we apply a truncation correction factor $\gamma$ to the measured
scattering coefficients ($b_{sca,Rayleigh}$), as discussed in Sect. 2.2.3 of the main text. We define $\gamma$ as the normalized ratio of the true
integrated scattering coefficient, $b_{sca,true}$ (i.e., what would be measured with an ideal integrating nephelometer) to the truncation-
affected scattering coefficient that is actually accessible to measurement by the instrument, $b_{sca,meas}$. Repeating Eq. (4) from
the main text for convenience,



$$\gamma = \frac{b_{sca,true}}{b_{sca,meas}} \cdot k_{Rayleigh} = \frac{b_{sca,true}}{b_{sca,meas}} \cdot \left( \frac{b_{sca,meas}^{Rayleigh}}{b_{sca,true}^{Rayleigh}} \right) \quad . \tag{A3}$$

The normalization factor $k_{Rayleigh}$ is required to represent the fact that some scattered light truncation is already implicitly accounted for in the CAPS PMssa cross calibration constant. For the recommended case of cross calibration with Rayleigh scatterers (i.e., Eq. 3), $k_{Rayleigh}$ represents the truncation of the Rayleigh scattered light from the calibration aerosol.

The coefficients $b_{sca,true}$ and $b_{sca,meas}$ in Eq. (A3) can be calculated using specific versions of the general Eq. (A1) for calculating

aerosol scattering coefficients $b_{sca}(\lambda)$. Following on from earlier integrating nephelometry studies (Anderson et al., 1996; Heintzenberg and Charlson, 1996; Moosmüller and Arnott, 2003; Müller et al., 2011b; Peñaloza M, 1999), we express this equation as a function of the scattering function of the particle population $S_p(\theta, \lambda)$, the light collection efficiency of the integrating sphere $\eta(\theta, \lambda)$, and the angular sensitivity function $Z(\theta)$ of the combined optical system:

$$b_{sca}(\lambda) = 2\pi \int_0^\pi S_p(\theta, \lambda) \eta(\theta, \lambda) Z(\theta) d\theta \quad . \tag{A4}$$


In this equation aerosol properties are represented by $S_p(\theta, \lambda)$ and instrument properties by $\eta(\theta, \lambda)$ and $Z(\theta)$ (which are assumed to be independent of the azimuthal scattering angle $\varphi$). Examples of each of these three functions are shown in Fig. 3 of the main text for the specific cases of $b_{sca,true}$ and $b_{sca,meas}$.

To calculate $S_p(\theta, \lambda)$, we assume that we are measuring an ensemble of spherical, homogeneous particles. In this case, $S_p(\theta, \lambda)$ is given by

$$S_p(\theta, \lambda) = \int_{-\infty}^{\infty} \frac{S_{11}(\theta, X, m, \lambda)}{X(D_p, \lambda)^2} \frac{\pi D_p^2}{4} \frac{dN}{dlogD_p} dlogD_p \quad , \tag{A5}$$

where $X$ is the particle size parameter ($= \pi D_p / \lambda$) and $S_{11}(\theta, X, m, \lambda)$ is the intensity-related scattering matrix element of a single

particle with size parameter $X$ and complex refractive index $m(\lambda)$ (Eq. 3.16 in Bohren and Huffman, 1998). For unpolarized incident light, like that used in the CAPS PMssa, $S_{11}$ describes the angular distribution of scattered light intensity. In this work we calculated $S_{11}$ with Mie theory (i.e., assuming homogenous spherical particles). Specifically, we calculated $S_{11}$ with modified version of Bohren and Huffman's Fortran routine (bhmie.f; Bohren and Huffman, 1998) by B.T. Draine (available at https://www.astro.princeton.edu/~draine/scattering.html, last access 14.05.2020). We further modified this routine to accept





programmatic inputs and outputs. Alternative forms of $S_p(\theta, \lambda)$ could be calculated with optical models that are not restricted to the assumption of spherical, homogenous particles, or $S_p(\theta, \lambda)$ could be measured directly with a polar nephelometer.

For an ideal integrating nephelometer, $Z(\theta) = \sin\theta$ (i.e., Eq. A2). Within the CAPS PMssa integrating sphere, no baffle is employed in front of the PMT to prevent the detection of directly scattered light (i.e. light that has not undergone any reflections

from the interior surface of the sphere). This could potentially lead to deviations from the ideal angular sensitivity condition. However, the fraction of the interior surface area of the sphere taken up by the PMT is only 0.6%, and calculations of the sphere properties indicate that the PMT light detection efficiency is independent of scattering angle to within 1% (Onasch et al., 2015). As a result, Onasch et al. (2015) assumed that the ideal condition of $Z(\theta) = \sin\theta$ is applicable for the CAPS PMssa. We make the same assumption here.


In other formulations of Eq. (A4), $Z(\theta)$ is typically expressed together with $\eta(\theta, \lambda)$ in a single function (e.g. Anderson et al. (1996) refer to this single function as '$f(\theta)$', while Müller et al. (2011) use '$Z_{ts}(\theta)$'). We reformulate this function as two separate components to highlight the importance of $\eta(\theta, \lambda)$, which describes the efficiency with which a given instrument can collect light. $\eta(\theta, \lambda)$ can vary between 0 (no light collected) and 1 (all light collected). Considering $\eta(\theta, \lambda)$ explicitly allows for

the simple and transparent introduction of additional physical processes into light scattering calculations (e.g. reflection from the glass tube containing the aerosol sample, as discussed below). Additionally, comparison of the $\eta(\theta, \lambda)$ curves of different instruments provides a clear and intuitive comparison of the abilities of the instruments to collect scattered light.

Within this formulation the notion that an ideal integrating nephelometer collects scattered light over all possible directions is

expressed by the condition $\eta_{ideal}(\theta, \lambda) = 1$ for $0 \leq \theta \leq \pi$ rad (as shown in Fig. 3). Scattered light truncation in real integrating nephelometers can be expressed by setting $\eta(\theta, \lambda)$ to 0 at angles where light is undetected. For example, for the specific case of the well-characterized TSI 3563 cell-direct integrating nephelometer (TSI Inc., St. Paul, MN, USA),

$$\eta_{TSI\_3563}(\theta, \lambda) = \begin{cases} 1 & \text{if } \theta_1 < \theta < \theta_2 \\ 0 & \text{if } 0 \leq \theta \leq \theta_1 \text{ or } \theta_2 \leq \theta \leq \pi \end{cases} \qquad \text{(A6)}$$

where $\theta_1 = 7°$ and $\theta_2 = 170°$ are referred to as the truncation angles of the instrument (Anderson and Ogren, 1998). For integrating sphere type nephelometers like the one used in the CAPS PMssa, truncation occurs because scattered light escapes through the aerosol sample entry and exit apertures in the sphere, as shown by the example truncation angles $\theta_1$ and $\theta_2$ in Fig. 1. In this case, the truncation angles depend on particle position along the longitudinal axis of the glass aerosol sample tube (Onasch et al., 2015), which is indicated as the $z$-dimension in Fig. 1. It is also possible for particles beyond the boundaries of

the integrating sphere to contribute to the measured scattering signal by scattering light into the sphere (Varma et al., 2003). In the CAPS PMssa, extra path lengths in the range from $l = 0$ to 4.7 cm outside the sphere boundaries must be considered.

The upper limit of this range is determined by the fixed positions of the aerosol flow tubing and ports in the optical cavity. Considering the instrument geometry shown in Fig. 1 the $z$-dependent truncation angles in the CAPS PMssa can be expressed as:


$$\theta_1(z) = \begin{cases} \tan^{-1}\left(\dfrac{d/2}{L/2 - z}\right) & \text{for } z \in \left[-(L/2 + l),\ L/2\right] \\[2ex] \pi + \tan^{-1}\left(\dfrac{d/2}{L/2 - z}\right) & \text{for } z \in \left[L/2,\ (L/2 + l)\right] \end{cases}$$

$$\theta_2(z) = \begin{cases} \pi + \tan^{-1}\left(\dfrac{d/2}{-L/2 - z}\right) & \text{for } z \in \left[-L/2,\ (L/2 + l)\right] \\[2ex] \tan^{-1}\left(\dfrac{d/2}{-L/2 - z}\right) & \text{for } z \in \left[-(L/2 + l),\ -L/2\right] \end{cases}$$

(A7)

Here it is assumed that the collimated light beam circulating in the optical cavity is confined along the central z-axis of the instrument with negligible width relative to the diameter of the glass sampling tube, and that multiple scattering effects from particles outside the collimated beam can be neglected.


Given these $z$-dependent truncations angles, and assuming that there is no scattered light reflection from the glass sampling tube ('no-refl'), the light collection efficiency function at a particular $z$-position (i.e., 'spot') in the CAPS PMssa optical cavity can be expressed as:

$$\eta\_spot_{meas}^{no-relf}(\theta, \lambda, z) = \begin{cases} 1 & \text{if } \theta_1(z) < \theta < \theta_2(z) \\ 0 & \text{if } 0 \leq \theta \leq \theta_1(z) \text{ or } \theta_2(z) \leq \theta \leq \pi \end{cases}$$

(A8)


The subscript 'meas' indicates that this is a function pertaining specifically to the CAPS PMssa (as opposed an ideal integrating nephelometer), following the notation in Eq. (A3).

In addition to highlighting the truncation angles, the $\eta$ formulation also allows explicit introduction of additional physical
processes that can reduce the probability of scattered light detection below 1 (which can be a function of both $z$ and $\theta$). In particular, we consider the process of reflection from the CAPS PMssa glass sampling tube (Fig. 1). This process is included in the radiative transfer theory model of Liu et al. (2018), but not the original scattered light truncation model presented by Onasch et al. (2015). To express this process in an $\eta$ function we first calculate the probability $R$ that light at an angle of




incidence of $\theta_i$ ($= \pi - \theta$) is reflected from the interface between the glass sampling tube and air (with refractive indices $m_{glass}$
and $m_{air}$, respectively) using the Fresnel equations for unpolarized incident light:

$$R(\theta_i, m_{glass}, m_{air}) = \frac{1}{2}\left(\left|\frac{m_{air}\cos\theta_i - m_{glass}\sqrt{1 - \left(\frac{m_{air}}{m_{glass}}\sin\theta_i\right)^2}}{m_{air}\cos\theta_i + m_{glass}\sqrt{1 - \left(\frac{m_{air}}{m_{glass}}\sin\theta_i\right)^2}}\right|^2\right.$$

$$\left.+ \left|\frac{m_{glass}\cos\theta_i - m_{air}\sqrt{1 - \left(\frac{m_{air}}{m_{glass}}\sin\theta_i\right)^2}}{m_{glass}\cos\theta_i + m_{air}\sqrt{1 - \left(\frac{m_{air}}{m_{glass}}\sin\theta_i\right)^2}}\right|^2\right). \tag{A9}$$

In the present study we assume $m_{glass} = 1.5 + 0i$ and $m_{air} = 1 + 0i$.

We calculate the number of reflections $N$ that light would need to undergo to exit the integrating sphere when scattered at an angle $\theta$ from a particle at position $z$ as the floor of the ratio of the distance of the particle from a sphere exit (along the $z$-dimension), to the $z$-component of the distance that the light would travel between each reflection event from the glass tube. An extra term is added to the numerator of this ratio to reflect the assumption that the particle lies along the center-line of the sampling tube:


$$N(z, \theta) = \begin{cases} \begin{cases} \left\lfloor\frac{\left|\frac{L}{2} - z + \left(\frac{d/2}{\tan\theta}\right)\right|}{d/\tan\theta}\right\rfloor & \text{if } z < L/2 \\ 0 & \text{if } z \geq L/2 \end{cases} & \text{if } \theta < \frac{\pi}{2} \\ \begin{cases} \left\lfloor\frac{\left|\frac{L}{2} + z + \left(\frac{d/2}{\tan(\pi-\theta)}\right)\right|}{d/\tan(\pi-\theta)}\right\rfloor & \text{if } z > -L/2 \\ 0 & \text{if } z \leq -L/2 \end{cases} & \text{if } \theta > \frac{\pi}{2} \end{cases} \tag{A10}$$





Equations (A9) and (A10) are calculated over the scattering angle range $0 \leq \theta \leq \pi$, and then combined to calculate a $z$- and $\theta$-dependent probability function that represents the total fraction of light reflected out of the integrating sphere. We term this function $R_{total}$:


$$R_{total}(z, \theta, m_{glass}, m_{air}) = R(\theta, m_{glass}, m_{air})^{N(z,\theta)} \quad . \tag{A11}$$

For a particular $z$-position in the optical cavity, $R_{total}$ can be combined with $\eta\_spot_{meas}^{no-refl}(\theta, \lambda, z)$ as defined in Eq. (A8) in order to calculate a light collection efficiency function that takes account of the process of glass tube reflection:

$$\eta\_spot_{meas}^{refl}(\theta, \lambda, z) = \eta\_spot_{meas}^{no-refl}(\theta, \lambda, z) \cdot (1 - R_{total}) \quad . \tag{A12}$$


Supplementary Fig. S16 displays example $R$, $N$, and $R_{total}$ curves as a function of scattering angle $\theta$ for the case of a particle in the center of the optical cavity ($z = 0$). Figure 3 displays example $\eta\_spot_{meas}^{refl}$ and $\eta\_spot_{meas}^{no-refl}$ curves (black and blue solid lines, respectively) for five different positions in the cavity spanning the range from $z = -6$ to 6 cm.

A single, integrated light collection efficiency curve $\eta(\theta, \lambda)$ can be generated for use in Eq. (A4) by integrating an $\eta\_spot$ function over all possible $z$-positions. That is,

$$\eta(\theta, \lambda) = \frac{1}{L + 2l} \int_{-(L/2+l)}^{L/2+l} \eta\_spot(\theta, \lambda, z) \, dz \quad . \tag{A13}$$

This operation assumes that the aerosol particles in the instrument are homogeneously distributed along the central $z$-axis of
the glass sampling tube, which is reasonable assumption to make for the aerosol number concentrations typically observed in the atmosphere (Qian et al., 2012). For concentrations much lower than this longer averaging times could be used to avoid any noise issues related to inhomogeneity. Example integrated light collection efficiency curves are displayed Fig. 3 for the cases where glass tube reflection is considered (black curve; 'with glass reflection') and is not considered (blue curve; 'no glass reflection'). In the terminology presented in this Appendix, these are the curves $\eta_{meas}^{refl}(\theta, m_{glass}, m_{air})$ and $\eta_{meas}^{no-refl}(\theta)$,
respectively, obtained by integrating the corresponding $\eta\_spot$ curves in Eq. (A13). The extra dependencies of $\eta_{meas}^{refl}$ on $m_{glass}$ and $m_{air}$ come from the dependence of this function on $R_{total}$ (Eq. A12).





All of the elements are now in place to use Eq. (A3) to calculate the truncation correction factor $\gamma$ for some aerosol with scattering phase function $S_p(\theta, \lambda)$. $b_{sca,true}$ is calculated by substituting $S_p(\theta, \lambda)$, $Z(\theta) = \sin \theta$, and $\eta(\theta, \lambda) = \eta_{ideal}(\theta, \lambda) = 1$ into

Eq. (A4). $b_{sca,meas}$ is calculated in the same manner, except $\eta(\theta, \lambda)$ is set to $\eta_{meas}^{refl}(\theta, m_{glass}, m_{air})$, if one wishes to account for the process of glass tube reflection, or $\eta_{meas}^{no-refl}(\theta)$ if one wishes to neglect this process. The normalization factor $k_{Rayleigh}$ is calculated in a similar way using the Rayleigh scattering phase function (Eq. 5.6 in Bohren and Huffman, 1998) and again assuming $Z(\theta) = \sin \theta$. Specifically,

$$k_{Rayleigh} = \frac{\int_0^\pi (1 + \cos^2 \theta) \sin \theta \, \eta_{meas}(\theta) \, d\theta}{\int_0^\pi (1 + \cos^2 \theta) \sin \theta \, \eta_{ideal}(\theta) \, d\theta} \quad . \tag{A14}$$


### 9. Appendix A2: Error model for the CAPS PMssa

To build an error model for the CAPS PMssa we begin with Eq. (2) from the main text, which is repeated here for convenience.

$$b_{abs} = b_{ext} - b_{sca} = \frac{1}{\alpha} \cdot \left( b_{ext,sample} - b_{ext,baseline} \right) - \frac{\gamma}{\beta} \cdot \left( b_{sca,sample} - b_{sca,baseline} \right) \quad . \tag{A15}$$

Individual uncertainty estimates for the seven parameters on the right hand side of this equation are given in Table 1. We assume that all of these errors are uncorrelated with each other. This is not generally true for the errors in the geometry correction factor $\alpha$ and the cross calibration constant $\beta$, since $\beta$ is directly proportional to $\alpha$ (Eq. 3; note that this is not the case for $\gamma$, since $\gamma$ is defined as a normalized ratio). However, in the specific case of a cross-calibrated instrument, we take the errors in $\beta$ to represent uncertainties arising from (i.e., precision) and after (i.e., drift) a cross calibration measurement. This part of

the overall uncertainty in $\beta$ is uncorrelated with the error in $\alpha$.

The error model is then constructed by applying the standard rules of error propagation to Eq. (A15), given the assumption of uncorrelated errors:

$$\delta b_{abs} = \sqrt{(\delta b_{ext})^2 + (\delta b_{sca})^2} \quad , \tag{A16}$$


where



$$\delta b_{ext} = b_{ext} \cdot \sqrt{\left(\frac{\delta \alpha}{\alpha}\right)^2 + \left(\frac{\sqrt{(\delta b_{ext,sample})^2 + (\delta b_{ext,baseline})^2}}{b_{ext,sample} - b_{ext,baseline}}\right)^2} \quad , \tag{A17}$$

and

$$\delta b_{sca} = b_{sca} \cdot \sqrt{\left(\frac{\delta \beta}{\beta}\right)^2 + \left(\frac{\delta \gamma}{\gamma}\right)^2 + \left(\frac{\sqrt{(\delta b_{sca,sample})^2 + (\delta b_{sca,baseline})^2}}{b_{sca,sample} - b_{sca,baseline}}\right)^2} \quad . \tag{A18}$$

The same quantities can also be used to calculate errors in SSA ($=b_{sca}/b_{ext}$) measured by the CAPS PMssa:

$$\frac{\delta SSA}{SSA} = \sqrt{\left(\frac{\delta b_{sca}}{b_{sca}}\right)^2 + \left(\frac{\delta b_{ext}}{b_{ext}}\right)^2} \quad . \tag{A19}$$

## 10. Data availability

Data archiving is currently underway. Data will be available on Zenodo if the manuscript is accepted for publication.

## 11. Competing interests

The authors declare that they have no conflicts of interest.

## 12. Author contributions

RLM, JCC, and MGB developed the error model based on the theoretical description of the instrument. RLM developed the new truncation model presented in Appendix A1 together with MGB. RLM, BB, MI, and MGB designed and/or performed the laboratory measurements of cross calibration constants and truncation values. RLM and FL calculated truncation values to compare with the laboratory measurements. RLM and MGB designed the Bologna mobile experiment. MB and REP took the measurements and analyzed the raw data during the Bologna campaign. JCC and TM took the measurements and analyzed the



raw data during the Melpitz campaign. JSH, MMM, KE, and MGB designed the Cabauw experiment. JSH coordinated the Cabauw campaign. RLM and PF took the measurements and analyzed the raw data during the Cabauw campaign. RLM performed the data analysis and interpretation and wrote the manuscript with input from JCC and MGB. All co-authors reviewed and commented on the manuscript.

**13. Acknowledgements and financial support**

We would like to thank Nicolas Bukowiecki, Birgit Wehner, Angela Marinoni, and Francisco Navas-Guzmán for their coordination support during the Melpitz, Bologna, and Payerne campaigns, and Martina Burger for her support during the laboratory experiments. This work was supported by the project "Metrology for light absorption by atmospheric aerosols", which is funded by the European Metrology Programme for Innovation and Research (EMPIR, grant no. "16ENV02 Black

Carbon"). The EMPIR initiative is co-funded by the EU Horizon 2020 research and innovation programme and the EMPIR participating states. The Swiss partners of this project are funded by the Swiss State Secretariat for Education, Research and Innovation (SERI; contract no. 17.00115, BlackC). JCC and FL were funded by Natural Resources Canada (EIP-EU-TR3-04A and TR3-3B03-0002B). The opinions expressed and arguments employed herein do not necessarily reflect the official views of the Swiss Government. RLM, MB, REP, and MGB received financial support from the ERC under grant ERC-CoG-615922-

BLACARAT. Trans-national access to the Cabauw, Melpitz and Bologna sites was supported by the ACTRIS–2 project (EU H2020–INFRAIA–2014–2015, grant agreement no. 654109).

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

**Table 1: Summary and description of uncertainties in the individual parameters comprising the error model described in Sect. 2.3.**
**The precision column represents uncertainty due to the limited precision with which a particular parameter can be determined during calibration or measurement, and the drift column represents uncertainty due to possible drift of a parameter between available measurements. Estimated values are taken from previous studies or this study as indicated. The estimated values with units of Mm$^{-1}$ correspond to absolute errors, and those with percentages relative errors.**

| Parameter | Symbol | Precision | Drift (stability-based uncertainty) | Description | References |
|---|---|---|---|---|---|
| Sample extinction coefficient | $b_{ext,sample}$ | 1 Mm$^{-1}$ | NA | Conservative estimate of short-term, random noise. | (Onasch et al., 2015) |
| Baseline extinction coefficient | $b_{ext,baseline}$ | 0.35 Mm$^{-1}$ | 0.3 Mm$^{-1}$ over 10 mins (CAPS630b) | Values estimated from the Cabauw field dataset (Fig. S8). | This study |
| Sample scattering coefficient | $b_{sca,sample}$ | 1 Mm$^{-1}$ | NA | Conservative estimate of short-term, random noise. | (Onasch et al., 2015) |
| Baseline scattering coefficient | $b_{sca,baseline}$ | 0.66 Mm$^{-1}$ | 0.1 Mm$^{-1}$ over 10 mins (CAPS630b) | Values estimated from the Cabauw field dataset (Fig. S8). | This study |
| Geometry correction factor | $\alpha$ | 1% | 3% over 1 year | Drift value determined from regular CAPS PMex measurements at the European Center for Aerosol Calibration | (Petzold et al., 2013; Pfeifer et al., 2020) |



| | | | | (ECAC; http://www.actris-ecac.eu/) | |
|---|---|---|---|---|---|
| Scattering cross calibration factor | $\beta$ | 2% (CAPS450) 2% (CAPS630a) 2% (CAPS630b) 6% (CAPS780) | NA (CAPS450) 2% (CAPS630a) 2.5% (CAPS630b) 8% (CAPS780) | Values estimated from the Payerne (Fig. 5) and Cabauw (Sect. 6.2.2) field datasets. | This study |
| Truncation correction factor | $\gamma$ | 4% for fine-mode dominated aerosol 9% for coarse-mode containing aerosol | NA | Values derived from the sensitivity analysis discussed in Sect. 6.2.3. | This study |


**Table 2: CAPS PMssa instrument units that were used in the present study**

| Unit ID | Wavelength (nm) | Institute | Serial number | Geometry correction factor ($\alpha$) |
|---|---|---|---|---|
| CAPS450 | 450 | PSI | 314003 | 0.78 |
| CAPS630a | 630 | PSI | 313004 | 0.71 |
| CAPS630b | 630 | Demokritos | 313003 | 0.7 – 0.73 |
| CAPS780 | 780 | PSI | 314002 | 0.78 |

**Table 3: Test aerosols used to measure scattered light truncation in the CAPS PMssa as a function of particle diameter and the values of parameters used in the corresponding model calculations. All particles were size classified by AAC.**





| Test aerosol | Refractive index (wavelength) | Geometric standard deviation of the size distributions | Dates of experiment repeats |
|---|---|---|---|
| PSL spheres | 1.59 + 0$i$ (at 450, 630 & 780 nm) | 1.05 and 1.1 | 16 to 22-08-2019; 16 to 20-01-2020 |
| DEHS (Di-Ethyl-Hexyl-Sebacat) | 1.46 + 0$i$ (at 450 nm); 1.45 + 0$i$ (at 630 & 780 nm) | 1.1 | 14 to 16-01-2019 |
| Ammonium sulphate | 1.50 + 0$i$ (at 450, 630 & 780 nm) | 1.1 | 26-04 to 08-05-2018; 16 to 30-08-2018 |


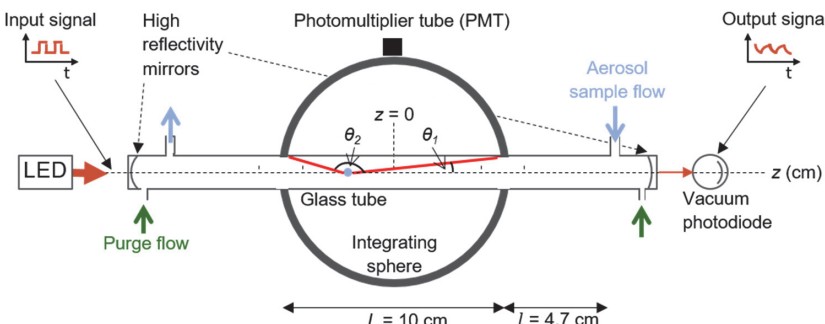

**Figure 1: Schematic diagram of the CAPS PMssa monitor with relevant components and variables highlighted. A glass tube encapsulates the aerosol sample to be measured. A light-emitting-diode (LED) delivers a square-wave modulated light signal as input to the optical cavity. The phase shift of the output signal from the cavity relative to the input signal is measured by a vacuum photodiode: this is the extinction channel of the instrument. Light scattered from the aerosol sample is collected by the integrating sphere and measured with a photomultiplier tube (PMT): this is the scattering channel of the instrument. $\theta_1$ and $\theta_2$ are the two truncation angles for light scattered from a particle at position $z$ along the instrument axis (without considering reflection from the glass tube).**







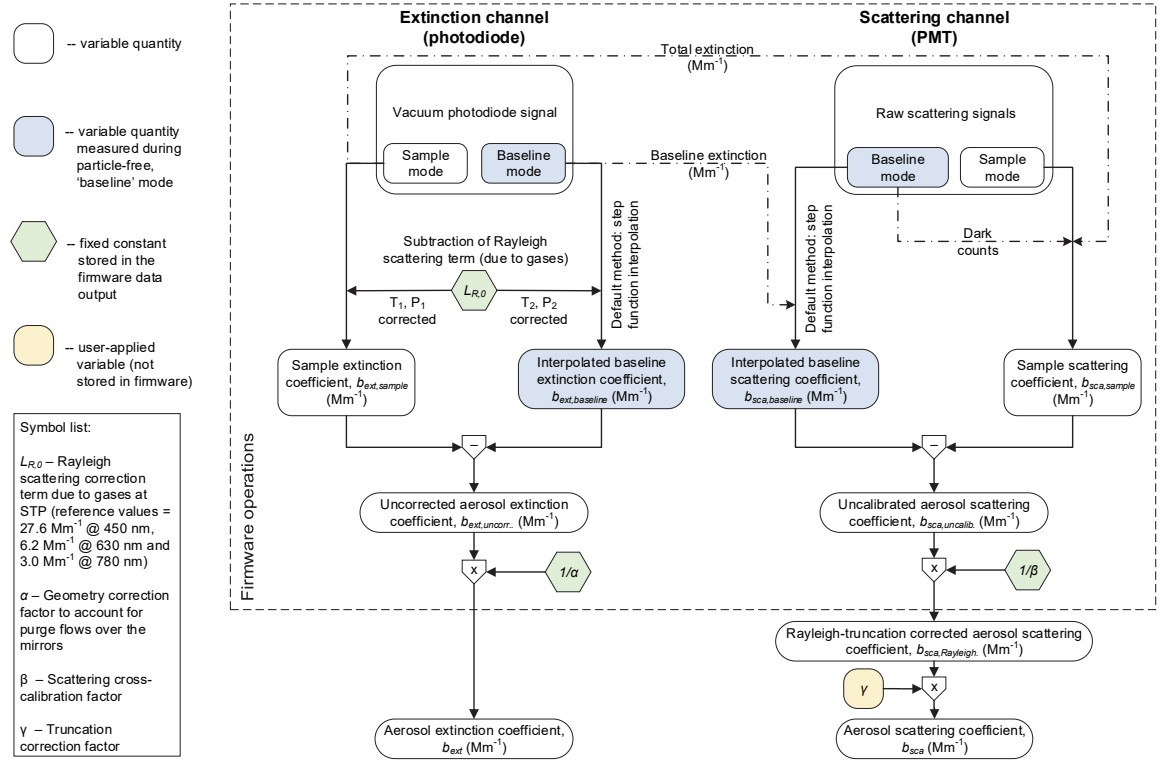

**Figure 2: Data processing chain for the extinction and scattering channels of the CAPS PMssa. Blue boxes indicate quantities that are measured during the periodic 'baseline' mode of operation of the instrument. Hexagonal containers indicate fixed constants, rounded rectangular containers represent variable quantities.**






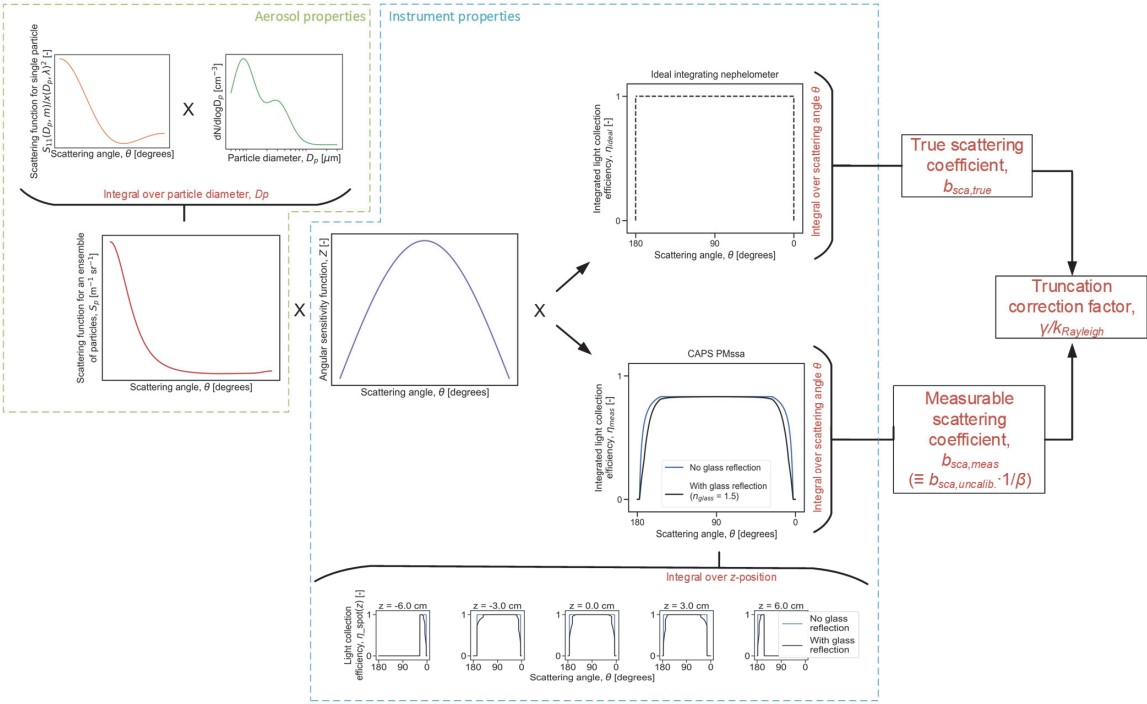

**Figure 3: Schematic diagram of the new model for calculating truncation correction factors for the CAPS PMssa. Full details of the calculations are presented in Appendix A1. The model requires as input a light collection efficiency function and an angular sensitivity function, which are determined by the geometry of the CAPS PMssa optical system; and a scattered light intensity function for the ensemble of particles being measured, which is a function of the particle size distribution (dN/dlogDp) and size-dependent aerosol scattering phase function. The main output of the model is the truncation correction factor, γ.**




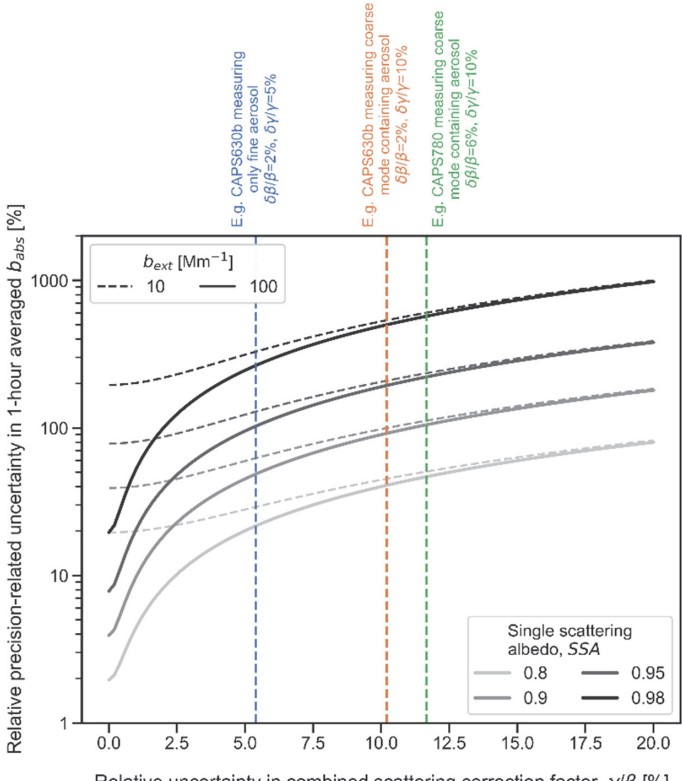

**Figure 4: Theoretically-calculated relative uncertainty in 1-hour averaged CAPS PMssa $b_{abs}$ measurements as a function of the**
**relative uncertainty in the combined scattering correction factor (defined in Eq. 6 using the ratio of the truncation correction factor**
**$\gamma$ and the instrument cross-calibration factor $\beta$). Curves are shown for four different SSA values (grey shading) and two different**
**aerosol loadings ($b_{ext}$ of 10 and 100 Mm⁻¹). The curves were generated using the error model presented in Sect. 2.3 and Appendix A2**
**with inputs that were chosen to represent instrument characteristics during the Cabauw field campaign, as detailed in the main text.**




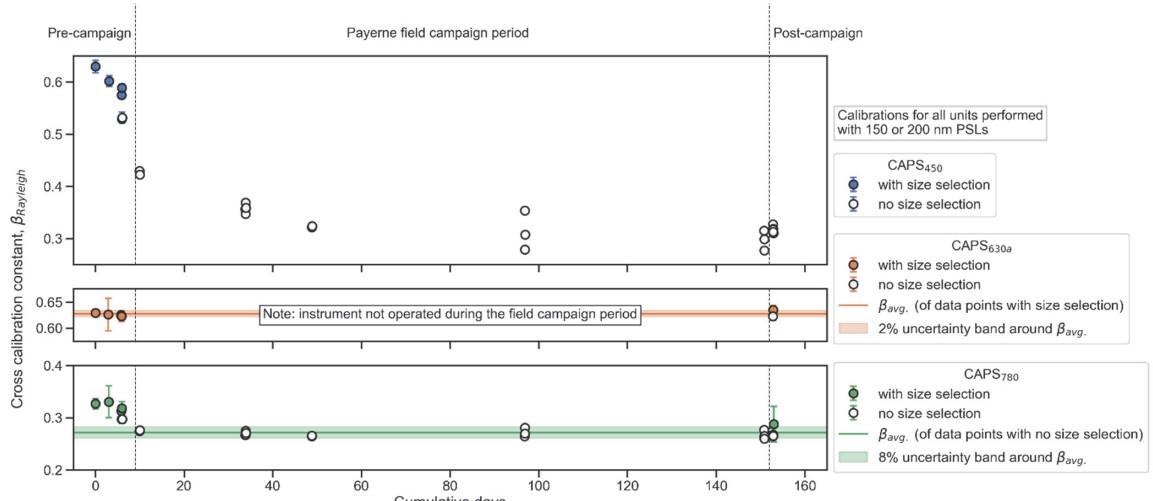


**Figure 5: Rayleigh-regime cross calibration constants ($\beta_{Rayleigh}$) for three CAPS PMssa units (CAPS450, CAPS630a, CAPS780) measured before, during and after the Payerne field campaign. Error bars indicate the standard deviation of the measured ratios used to determine each $\beta_{Rayleigh}$ value (see Sect. 3.2).**


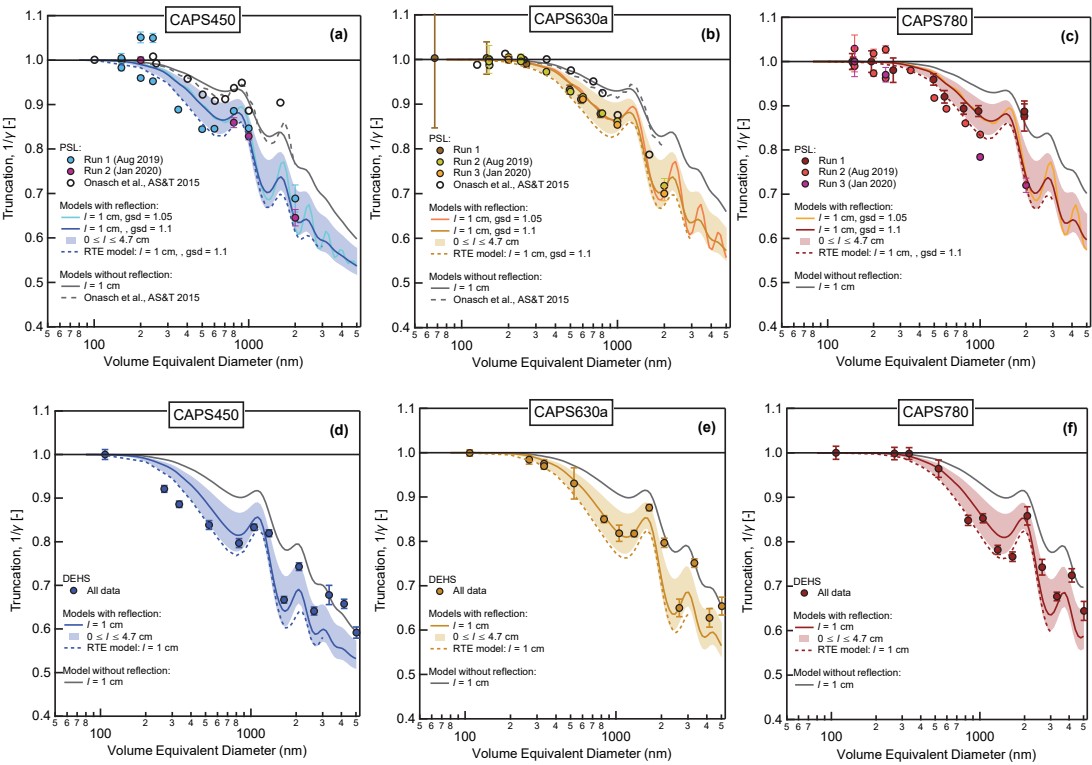

**Figure 6: Measured and modeled truncation values as a function of volume equivalent particle diameter for PSL and DEHS aerosols. The truncation values plotted on the y-axes correspond to the inverse of the truncation correction factor γ defined by Eq. (4). Modeled curves were calculated with the truncation model presented in Appendix 1 as well as the radiative transfer equation (RTE) model**
**presented by Liu et al. (2018). The parameter *l* represents the extra path length beyond the integrating sphere, and gsd refers to the geometric standard deviation of the modeled test aerosols.**

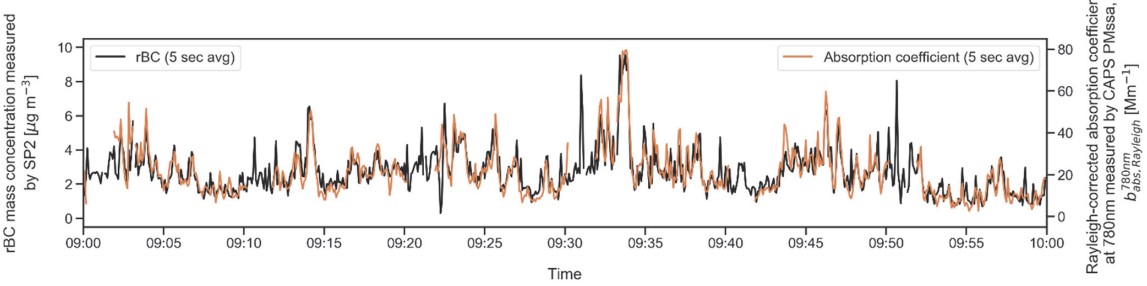

**Figure 7: Time series of a 1-hour period of 5-sec averaged rBC mass concentration and absorption coefficient measurements made**
**from the mobile laboratory traveling on a highway road during the Bologna campaign. rBC mass concentrations were measured by**





SP2 – a sensitive, single-particle-based instrument. Absorption coefficients were measured at 780 nm with the unit CAPS780. Note that the absorption coefficients have only been corrected for Rayleigh scattering truncation and therefore should not be interpreted quantitatively (e.g. they cannot be used to calculate MAC of BC values). Spikes in the time series correspond to emissions from passing vehicles on the highway.

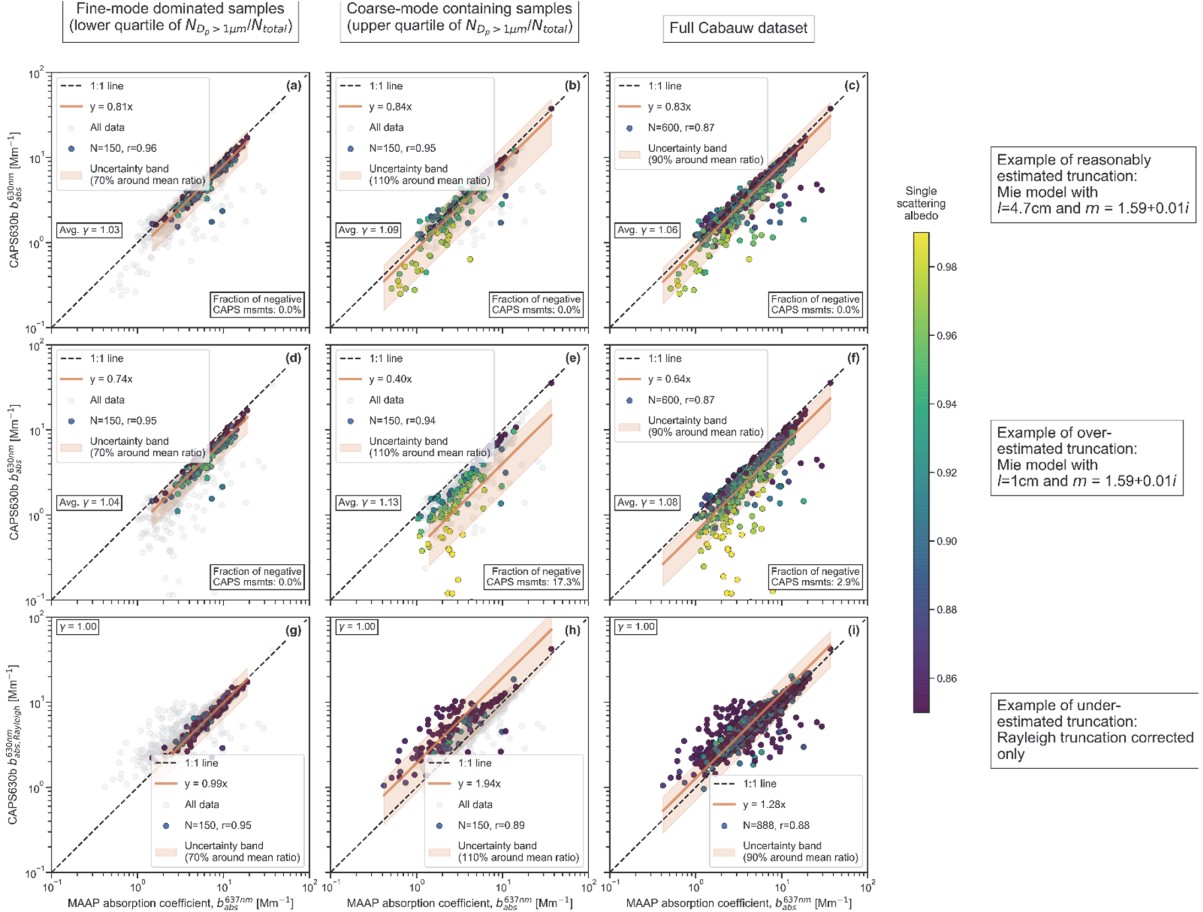


**Figure 8:** Comparison of CAPS PMssa absorption coefficients at 630 nm ($b_{abs}^{630nm}$) and MAAP absorption coefficients at 637 nm ($b_{abs}^{637nm}$) for three different truncation correction scenarios. The three columns represent different subsets of the dataset, as indicated in the column titles. The three rows represent the different truncation correction scenarios, as indicated in the row labels. The top row (panels a-c) display CAPS PMssa measurements that were corrected using the Mie-based truncation correction scheme presented in Appendix A1 with $m$ set to 1.59+0.01$i$ and $l$ set to 4.7 cm. The middle row (panels d-f) displays the same type of data with $l$ set to 1 cm. The bottom row (panels g-i) contains CAPS PMssa measurements corrected for the truncation of Rayleigh scattering only (i.e., $\gamma = 1$). All data points are colored by the SSA values calculated from the same CAPS PMssa data shown in each panel. The mean ratios of CAPS PMssa to MAAP $b_{abs}$ are plotted as solid orange lines in each panel. The uncertainty bands represent the 95th percentiles of theoretical uncertainties calculated with the error model and inputs described in Sect. 2.3, with relative uncertainties in $\gamma$ of 4, 9, and 6% for the fine-mode dominated, coarse-mode containing, and full dataset, respectively.