# Peer review of "Detailed characterization of the CAPS single scattering albedo monitor (CAPS PMssa) as a field-deployable instrument for measuring aerosol light absorption with the extinction-minusscattering method"

_Atmospheric Measurement Techniques, 2020_

## Referee Comment (RC1) · Timothy Onasch (Referee) · 15 Sep 2020

Review of Manuscript

Title: **Detailed characterization of the CAPS single scattering albedo monitor (CAPS PMssa) as a field-deployable instrument for measuring aerosol light absorption with the extinction-minus scattering method**

Authors: R. L. Modini et al.

**Overview:** This paper investigates the errors associated with the extinction, scattering, and absorption measurements of the recently commercialized CAPS PMssa instrument on deployment as a field-based instrument. The paper focuses on the method and errors associated with the scattering truncation and scattering calibration (to the extinction measurement) and how these relate to the absorption measurement. Field data cases are shown that highlight the strengths and weaknesses of the CAPS PMssa absorption measurement. The authors build a truncation model for the CAPS PMssa that explicitly includes the glass tube inside the scattering sphere and build a full error model for the EMS-based (Extinction Minus Scattering) absorption measurement. Finally, a list of recommendations is given to ensure accurate absorption measurements in the field using the CAPS PMssa instrument. I find this manuscript well written and organized. In general, I have very little argument with their structure, methods, and conclusions. This paper has a relevant and timely topic for the Atmospheric Measurement Techniques (AMT) journal, specifically, and the atmospheric community, in general. In particular, the manuscript addresses the critical need for field-based absorption measurements on freely-floating particles (i.e., not filter-based measurements). This article deserves to be published in AMT and is almost ready for direct publication. I have a few specific comments that should be addressed prior to publication.

1.) Terminology: Filter-based instruments do not measure absorption. In general, the authors are very careful about this issue; however, it is worth restating. First sentence on the last paragraph of page 2, for example, states: "aerosol light absorption has been measured by detecting the attenuation of light transmitted through aerosol samples deposited on filter substrates." Perhaps a slightly more accurate portrayal would be to state: "aerosol light absorption derived by measuring the attenuation of light transmitted through aerosol samples deposited on filter substrates."

2.) The Multi-Angle Absorption Photometer (MAAP) instrument is highlighted by the authors as an improved filter-based method for deriving aerosol light absorption from light attenuation and scattering from an aerosol loaded filter substrate. The MAAP is indeed an impressive filter-based system; however, it is still a filter-based attenuation (and scattering) technique and therefore susceptible to filter substrate-based issues, in addition to potential failures in the 2 stream radiative transfer model approximation from which the absorption is derived. The MAAP is not an absorption standard in any sense and this should be acknowledged. The authors describe some of the errors associated with the MAAP on page 16, but do not include this in the analysis of the instrument comparison in section 6.2. It is understandable that these errors are not emphasized in the comparison, as the focus of the work is on the CAPS PMssa absorption measurement. However, in this case the authors should either state that their analysis of this data make the implicit assumption that the MAAP data has no uncertainties for this comparison, or acknowledge the inherent errors in the MAAP by including them in the comparison. For example, all of the differences between the CAPS PMssa absorption and the MAAP absorption

measurements are all explained as potential issues with the CAPS, but no discussion is given about potential issues with the MAAP measurements.

3.) On page 6 line 175, the authors state that the mirror purge flow is "drawn" continuously over the mirrors, whereas the purge flow is actually pushed (i.e., positive flow).

4.) On page 6 line 182, the authors state that the extinction channel signal is monitored by a vacuum photodiode.  The one exception for this is the 780nm wavelength system which uses a PMT detector for greater sensitivity at the longer wavelengths.

5.) Page 7 line 221 and Figure 2, the scattering channel is normalized to the amount of light circulating in the optical cell during the LED off phase using the reported "signal" levels and is a function of the cell loss, which controls the rate of decay of light in the cell.  Thus, the authors are incorrect when they note that the total extinction is used in the scattering channel ($b_{sca,sample}$) calculation.  Extinction is related to the loss, but is itself a differential measurement of the loss measured with sample from the loss measured without sample.  This needs to be corrected.

6.) Page 9 line 260 (and elsewhere), the authors note that the CAPS PMssa firmware automatically applies a geometry correction factor of 0.7.  The manufacturer's geometry correction factor, as noted in Onasch et al. (2015), is 0.73 for CAPS PMssa monitors.  It has been noted that early instruments may have dropped the second digit in the data files, even though the internal value was always 0.73.

7.) Page 9 line 273, "N2" should read "NO2"

8.) Page 9 line 275, the authors state "we have observed that the instrument can take a long time (~hours) to adjust and stabilize when filled with different gases."  Note that this may be true for the CAPS PMssa monitor due to the purge flows that have large area filters and very slow flows, but is not true of the CAPS NO2 systems, for example, that do not have the more complicated flows.  Thus, this sentence should be modified to reflect the actual or presumed reason for this observed behavior.  There is nothing mysterious about this issue.  Further, this issue will be more of an issue for "sticky" gases, such as water vapor.

9.) Page 15 line 468, remove "anyway" from "anyway small relative"

10.) Page 19 line 597, the authors write "A number of repeat experiments were performed for some of the aerosol types, as indicated in Table 3." Table 3 lists the "Dates of experimental repeats". That said, the actual dates are not the important information here, rather the number of measurements (i.e., repeats) and the time between is the useful information. Suggest modifying Table 3 to address.

11.) Table 3 – why did the authors chose to use 1.5+0i for ammonium sulfate refractive index?  Toon et al. 1976 suggests a value closer to 1.53 for VIS.  Presumably this minor difference makes little to no effect.

12.) Page 22 lines 676 to 688 and figure S6.  In Figure S6 and in this paragraph, the authors discuss an issue they observed in a field study that is not discussed in the paper except for this single example. The example shown comes from a field study where the highly reflective mirrors on the CAPS PMssa monitor deployed became significantly contaminated.  Under these conditions, the instrument is operating outside of its intended parameter space, though to be fair, the intended parameter space on dirty mirrors is not defined by a set threshold, but rather by operator decision making.  The data in the figure and discussed in the text notes that as the mirrors get dirtier, the extinction measurement increases relative to a CAPS PMex system and a

nephelometer.  As the extinction channels in the CAPS PMssa and the CAPS PMex are nearly identical (SSA slightly shorter than EX), the increase in the mirror contamination could have readily happened in either system.  In other words, there is nothing here specific to the CAPS PMssa compared with the CAPS PMex for the extinction measurement.  The authors discuss this issue as if it were related to a non-linearity at high extinction levels, though later decide that this does not fit the observations.  They do not note that the mirrors become dirty when either (a) there is a significant pressure burst that can lift sediment from walls and deposit on the mirrors or (b) the purge system in the CAPS PMssa is failing, causing particles to access the purge regions.  The latter explains both the increase in the baseline loss of the cell due to dirty mirrors and the observed increase in the extinction coefficient as the sample pathlength increases as particles cross the light beam inside the purge regions that are normally particle free (i.e., the geometry parameter decreases).  The authors conclude that they (i.e., the operators) must be vigilant and keep the mirrors clean.  In essence, this paragraph and supplemental figure are hardly relevant to the current discussion of errors in the CAPS PMssa instruments measure of particulate absorption, as purge failure is a potential issue for any CAPS PMssa or PMex system.  At worse, this section/figure could cause readers to think that the CAPS PMssa fails under high loadings, which is not true.  Basically, while this section/figure aren't incorrect, it does not add anything useful to the discussion of errors at hand and could potentially cause confusion.  I would suggest this section/figure be removed, or at least, qualified by noting that the issue here were likely caused by a purge system failure that went undetected during a field study, which is not a minor contamination issue, and under such out-of-tolerance conditions the CAPS PMssa, like any instrument, struggles to meet specifications.  For example, the authors use this information to make the statement, "the CAPS method is effectively 'calibration free' (apart from the geometry correction factor, as discussed in Sect. 2.2.1, as well as potential non-linearities at high baseline losses)" on page 9.  I would suggest that this over emphasizes an issue that was caused by user fault (failure of purge flow), rather than potential measurement fault.

13.) Page 10 line 297 add "to" in "… we refer **to** the calibrated scattering coefficient…"

14.) Coarse mode.  The authors include significant discussions in this paper relating to the errors in b_sca and b_abs by the CAPS instrument (Sections 6.2.2, 6.2.3) and during the instrument correlation with the MAAP in section 6.2.4 when measuring coarse mode (i.e., PM10) particles.  While these discussions are important, the authors need to qualify these discussions with the facts that neither the CAPS nor the MAAP have been extensively tested with coarse mode particles (to my knowledge).  For example, the discussion of errors in the MAAP on page 16 mention the RAOS study and the Mueller et al. (2011a) study, but neither of these studies included coarse mode particles (i.e., particles greater than 1 micron in diameter).  Thus, while including these discussions here is important and relevant to the ambient results, the authors need to acknowledge that neither technique has been extensively studied for coarse mode particles (to my knowledge), which limits the quantitative aspect of these discussions.  Furthermore, a large fraction of the discussions on Figure 8 and the subsequent analysis relies on the coarse-mode and full dataset (which is highly biased by a coarse-mode).  The authors should at least qualify these results with respect to the lack of coarse mode quantification in both instrument techniques.

15.) Page 25 line 765, very nice correlation for high time resolution data. What does the correlation plot (or histogram of the ratios) of CAPS PMssa b_abs to SP2_rBC look like for the data presented in Figure 7? Since you explicitly state not to look at the quantitative value, I did that. The apparent ratio is 8 m2/g, which is very high for a wavelength of 780nm. What are the SSA values for these samples? The authors point out that the one unknown (i.e., not measured) component during this study is the scattering truncation (i.e., size distributions). For scattering truncation uncertainties, the SSA would have to be high to attempt to account for these apparently high MAC values. What were the size distributions of rBC measured by the SP2? What about the scattering measurements by SP2? Could the SP2 be saturating at these high rBC levels? What about absorption enhancement due to coated rBC particles (would be ~1.5x assuming a 1/wavelength from 550nm 7.5 m2/g MAC, which might not be unreasonable)? Perhaps, it would be either better to (a) remove one or both axes if considered non-quantitative or (b) provide a potential rationale for this apparent issue.

16.) Section 6.2. This section analyzes a comparison of the CAPS absorption measurements compared with the MAAP absorption measurements. As noted above in points 2 and 14, both instruments have associated errors for submicron and potential unknown errors for supermicron. The authors state (page 25, line 784) that this was done to "perform a full quantitative assessment of the ability of the CAPS PMssa to measure absolute aerosol absorption coefficients…". In reality, this section is a direct instrument to instrument comparison, rather than a quantitative assessment for the measurement of absolute aerosol absorption coefficients. It is likely that the errors in the MAAP are smaller than the errors in the CAPS b_abs, but they should still be included. What MAC value was assumed for the MAAP to derive the b_abs values? On page 28, when discussing the ~20% discrepancy between the CAPS b_abs and MAAP b_abs, the only errors discussed are those of the CAPS geometry correction factor and truncation factor, which are only on the order of 2-9%. The MAAP uncertainties stated on page 16 indicate similar orders of magnitude (5-7%), but are not discussed.

17.) Figure 8. This figure shows a lot of useful information tightly packed into one figure. The problems of using log-log plots for instrument intercomparisons is that it (a) blows up the noise at low signals and (b) removes any potential negative values from the comparisons. My first question here is how were the fits done for this figure? For example, the fit in the center plot does not go through the data shown at all. Thus, it is not a linear fit of the data in linear space and placed on log-log axes. Further, the fits assume that there is no y-intercept (or x-intercept). If it is a power law fit to the data, then there must be a lot of data not shown that is highly biasing the power fit. How do the fits account for the negative values? More information is warranted on the fitting approach and why it was chosen.

18.) Figure S15 caption indicates that the fit lines in Figure 8 are not fits, but rather mean ratios…. – this information needs to get into the main text and main figure captions!!! By taking the mean ratio, the resulting values are heavily influenced by noise in the smaller measurements. This would explain why the orange lines do not pass through the majority of the data points, especially at higher values of b_abs and lower values of SSA, where one would expect better agreement. Another potential approach that could be taken would be to estimate a conservative limit (for example 50 Mm-1 was chosen for this purpose when looking at the geometry correction factors above) and then look at the histograms of the ratios (again similar to the geometry correction factor analysis). This approach might better define the mean,

variance, and skewness of the resulting histograms, where the skewness could be used as a measure of how well the truncation models are being applied, rather than the vague (though true) color trends.

19.) Equation A11 takes the probably, R, which is an average of the s and p polarizations (see Eq A9), and takes it to the power of the number of estimated reflections for a light ray to exit the glass tube. As s and p polarizations have significantly different reflection probabilities as a function of incident angle, equation A11 is an approximation. Ideally, one would calculate $R\_s$ and $R\_p$ separately for the two polarizations and then average. Not sure how much of a difference this would make, but it would be more accurate.

---

## Referee Comment (RC2) · Rob L. Modini et al. · 16 Sep 2020

This study provides a detailed characterization and thorough discussion on the performance of commercialized CAPS PMssa in determining aerosol light absorption using the extinction minus scattering method and the associated uncertainties with this method. While the paper is well written in many aspects, I am a bit concerned that the key points and many useful information may be missed by the readers because of the extraordinary length of the manuscript. In general, I feel that Section 2 and 3 can be combined and shortened to make the experimental method clearer and to the

point. Specifically, in section 2.1 and 2.2, Onasch (2015) was heavily cited and many of the description of the instrument is repetitive. Section 2.3 and the sub-sections of Sect 2.2 are also mixed with extensive discussion, which can either to be moved to later sections or to Appendix. Also, the results from the Bologna campaign seems less relevant to the main focus of this study: identifying and quantifying uncertainties in the absorption measurement due to the scattered light truncation effect and cross calibration constant. I recommend publication of this manuscript after the above points are considered.

---

## Author Comment (AC1) · 13 Nov 2020

We thank the editor and reviewers for the timely handling of our manuscript, particularly during this difficult pandemic year. Please find below our point-by-point responses to each of the reviewer comments, including descriptions of the modifications we have made (or plan to make) to the manuscript. Reviewer comments are in black text and our responses are given in blue text.

**Referee #1: Timothy Onasch**

**General comments:**

Overview: This paper investigates the errors associated with the extinction, scattering, and absorption measurements of the recently commercialized CAPS PMssa instrument on deployment as a field-based instrument. The paper focuses on the method and errors associated with the scattering truncation and scattering calibration (to the extinction measurement) and how these relate to the absorption measurement. Field data cases are shown that highlight the strengths and weaknesses of the CAPS PMssa absorption measurement. The authors build a truncation model for the CAPS PMssa that explicitly includes the glass tube inside the scattering sphere and build a full error model for the EMS-based (Extinction Minus Scattering) absorption measurement. Finally, a list of recommendations is given to ensure accurate absorption measurements in the field using the CAPS PMssa instrument. I find this manuscript well written and organized. In general, I have very little argument with their structure, methods, and conclusions. This paper has a relevant and timely topic for the Atmospheric Measurement Techniques (AMT) journal, specifically, and the atmospheric community, in general. In particular, the manuscript addresses the critical need for field-based absorption measurements on freely-floating particles (i.e., not filter-based measurements). This article deserves to be published in AMT and is almost ready for direct publication. I have a few specific comments that should be addressed prior to publication.

We thank Dr. Onasch for his careful review and constructive comments. These have helped to improve the paper by clarifying some important points. We have addressed each of the comments as explained in the detailed responses below.

**Specific comments:**

1.) Terminology: Filter-based instruments do not measure absorption. In general, the authors are very careful about this issue; however, it is worth restating. First sentence on the last paragraph of page 2, for example, states: "aerosol light absorption has been measured by detecting the attenuation of light transmitted through aerosol samples deposited on filter substrates." Perhaps a slightly more accurate portrayal would be to state: "aerosol light absorption derived by measuring the attenuation of light transmitted through aerosol samples deposited on filter substrates."

Agreed. We have modified the sentence in question as suggested. It now reads: "Traditionally, aerosol light absorption has been derived by measuring the attenuation of light transmitted through aerosol samples deposited on filter substrates (e.g. Rosen et al., 1978)"

2.) The Multi-Angle Absorption Photometer (MAAP) instrument is highlighted by the authors as an improved filter-based method for deriving aerosol light absorption from light attenuation and scattering from an aerosol loaded filter substrate. The MAAP is indeed an impressive filter-based system; however, it is still a filter-based attenuation (and scattering) technique and therefore susceptible to filter substrate-based issues, in addition to potential failures in the 2 stream radiative transfer model approximation from which the absorption is derived. The MAAP is not an absorption standard in any sense and this should be acknowledged. The authors describe some of the errors associated with the MAAP on page 16, but do not include this in the analysis of the instrument comparison in section 6.2. It is understandable that these errors are not emphasized in the comparison, as the focus of the work is on the CAPS PMssa absorption measurement. However, in this case the authors should either state that their analysis of this data make the implicit assumption that the MAAP data has no uncertainties for this comparison, or acknowledge the inherent errors in the MAAP by including them in the comparison. For example, all of the differences between the CAPS PMssa absorption and the MAAP absorption measurements are all explained as potential issues with the CAPS, but no discussion is given about potential issues with the MAAP measurements..

We agree the MAAP should not be considered as a true absorption standard. Our intention is rather to use the MAAP as a common reference point, since the instrument unit-to-unit variability for the technique has proven to be very good. Thus, by comparing our CAPS PMssa absorption coefficients against those measured by a MAAP we expect future studies will also be able to use MAAP measurements as a way to link to the CAPS PMssa results presented in this work.

We also agree that these points should be stated more explicitly in the main text. We have added additional explanation to Sect. 3.1.1 and Sect. 6.2 (also in line with reviewer comments 14 and 16) in order to highlight that we do not consider the MAAP measurements to be a true absorption standard, to mention that its ability to measure super-micrometer particles requires further investigation, and finally to include discussion of MAAP errors alongside the CAPS PMssa error discussions.

3.) On page 6 line 175, the authors state that the mirror purge flow is "drawn" continuously over the mirrors, whereas the purge flow is actually pushed (i.e., positive flow)..

Thanks for the clarification. We have changed 'drawn' to 'pushed'.

4.) On page 6 line 182, the authors state that the extinction channel signal is monitored by a vacuum photodiode. The one exception for this is the 780nm wavelength system which uses a PMT detector for greater sensitivity at the longer wavelengths.

Thanks for the clarification. We have changed the sentence accordingly ("…is monitored by a vacuum photodiode or, in the case of the 780 nm unit, a photomultiplier tube (PMT)").

5.) Page 7 line 221 and Figure 2, the scattering channel is normalized to the amount of light circulating in the optical cell during the LED off phase using the reported "signal" levels and is a function of the cell loss, which controls the rate of decay of light in the cell. Thus, the authors are incorrect when they note

that the total extinction is used in the scattering channel (bsca,sample) calculation. Extinction is related to the loss, but is itself a differential measurement of the loss measured with sample from the loss measured without sample. This needs to be corrected..

Thanks for this clarification. We have changed the relevant labels in Fig. 2 from 'extinction' to 'optical loss'.

6.) Page 9 line 260 (and elsewhere), the authors note that the CAPS PMssa firmware automatically applies a geometry correction factor of 0.7. The manufacturer's geometry correction factor, as noted in Onasch et al. (2015), is 0.73 for CAPS PMssa monitors. It has been noted that early instruments may have dropped the second digit in the data files, even though the internal value was always 0.73.

Thanks for this clarification. We indeed took the figure of 0.7 from the header of the data acquisition files where the 2nd digit has been dropped. We have changed the text in all relevant places from 0.7 to 0.73, in order to represent the actual internal value. It was also necessary to reprocess the data displayed in Figs. 4 and 8 using the value of 0.73 rather than 0.7. As is to be expected with such a small change, the resulting changes to the final plots are very minor and do not alter our final conclusions.

7.) Page 9 line 273, "N2" should read "NO2"

The 'N2' was deliberate and meant to refer to the low span calibration point. But we agree that that doesn't really fit with the sentence and causes confusion. We have simply deleted the N2 to prevent any possible confusion.

8.) Page 9 line 275, the authors state "we have observed that the instrument can take a long time (~hours) to adjust and stabilize when filled with different gases." Note that this may be true for the CAPS PMssa monitor due to the purge flows that have large area filters and very slow flows, but is not true of the CAPS NO2 systems, for example, that do not have the more complicated flows. Thus, this sentence should be modified to reflect the actual or presumed reason for this observed behavior. There is nothing mysterious about this issue. Further, this issue will be more of an issue for "sticky" gases, such as water vapor.

We have added a qualifier to the end of the sentence to indicate that this behavior is expected. It now reads: "…we have observed that the instrument can take a long time (~hours) to adjust and stabilize when filled with different gases (as expected due to the low flows and large filter areas in the purge flow setup)"

9.) Page 15 line 468, remove "anyway" from "anyway small relative"

The "anyway" has been removed as suggested.

10.) Page 19 line 597, the authors write "A number of repeat experiments were performed for some of the aerosol types, as indicated in Table 3." Table 3 lists the "Dates of experimental repeats". That said, the actual dates are not the important information here, rather the number of measurements (i.e., repeats) and the time between is the useful information. Suggest modifying Table 3 to address.

Agreed. We will modify the table so that the number of repeat measurements can be easily determined.

11.) Table 3 – why did the authors chose to use 1.5+0i for ammonium sulfate refractive index? Toon et al. 1976 suggests a value closer to 1.53 for VIS. Presumably this minor difference makes little to no effect.

This was a typo, thanks for picking this up. The ammonium sulphate calculations were actually performed with a refractive index of 1.52, in line with a number of chemical information databases and closer to the Toon et al. value. We have modified Table 3 to fix this typo.

12.) Page 22 lines 676 to 688 and figure S6. In Figure S6 and in this paragraph, the authors discuss an issue they observed in a field study that is not discussed in the paper except for this single example. The example shown comes from a field study where the highly reflective mirrors on the CAPS PMssa monitor deployed became significantly contaminated. Under these conditions, the instrument is operating outside of its intended parameter space, though to be fair, the intended parameter space on dirty mirrors is not defined by a set threshold, but rather by operator decision making. The data in the figure and discussed in the text notes that as the mirrors get dirtier, the extinction measurement increases relative to a CAPS PMex system and a nephelometer. As the extinction channels in the CAPS PMssa and the CAPS PMex are nearly identical (SSA slightly shorter than EX), the increase in the mirror contamination could have readily happened in either system. In other words, there is nothing here specific to the CAPS PMssa compared with the CAPS PMex for the extinction measurement. The authors discuss this issue as if it were related to a non-linearity at high extinction levels, though later decide that this does not fit the observations. They do not note that the mirrors become dirty when either (a) there is a significant pressure burst that can lift sediment from walls and deposit on the mirrors or (b) the purge system in the CAPS PMssa is failing, causing particles to access the purge regions. The latter explains both the increase in the baseline loss of the cell due to dirty mirrors and the observed increase in the extinction coefficient as the sample pathlength increases as particles cross the light beam inside the purge regions that are normally particle free (i.e., the geometry parameter decreases). The authors conclude that they (i.e., the operators) must be vigilant and keep the mirrors clean. In essence, this paragraph and supplemental figure are hardly relevant to the current discussion of errors in the CAPS PMssa instruments measure of particulate absorption, as purge failure is a potential issue for any CAPS PMssa or PMex system. At worse, this section/figure could cause readers to think that the CAPS PMssa fails under high loadings, which is not true. Basically, while this section/figure aren't incorrect, it does not add anything useful to the discussion of errors at hand and could potentially cause confusion. I would suggest this section/figure be removed, or at least, qualified by noting that the issue here were likely caused by a purge system failure that went undetected during a field study, which is not a minor contamination issue, and under such out-of-tolerance conditions the CAPS PMssa, like any instrument, struggles to meet specifications. For example, the authors use this information to make the statement,

"the CAPS method is effectively 'calibration free' (apart from the geometry correction factor, as discussed in Sect. 2.2.1, as well as potential non-linearities at high baseline losses)" on page 9. I would suggest that this over emphasizes an issue that was caused by user fault (failure of purge flow), rather than potential measurement fault.

While we agree that this example is somewhat separated from the rest of the paper we think it is important to show for the following reasons. 1) As noted there is no set threshold above which a user can be sure mirror contamination is affecting the measurements. Although this will vary from unit to unit, we still think it is useful to show an example of the behavior that might be expected for these types of increases in the baseline optical loss (we are often asked this type of question by new users). 2) The fact that the extinction measurement became biased as the contamination increased while the scattering measurement remained more stable implies the instrument's cross calibration constant varied due to the contamination event. In this sense, we think the result is important to show and fits nicely within the context of the section in question, Sect. 4, which is focused on the stability of the cross calibration constant over time.

Having said that, we agree that more emphasis should be placed on qualifying the fact that this is an example of when the instrument is operating outside of its intended operation range, as well as the reasons this could occur (purge flow failure or pressure burst). We will modify the revised manuscript accordingly as suggested. We will also explicitly state that this type of instrument failure isn't directly related to measuring high aerosol loads to try and prevent readers drawing this incorrect conclusion.

Finally, we will remove the two sentences on L684 of the AMTD version of the manuscript relating to the non-linear regime hypothesis and then the ruling out of this hypothesis. However, we note that this example was not the only reason behind the statement on L267 about "…potential non-linearities at high baseline losses)." We have observed this type of behavior in laboratory experiments with both CAPS PMssa and PMex systems. Therefore, we elect to keep this particular statement as is.

13.) Page 10 line 297 add "to" in "… we refer to the calibrated scattering coefficient…"

Thanks for picking this up. The "to" has been added.

14.) Coarse mode. The authors include significant discussions in this paper relating to the errors in b_sca and b_abs by the CAPS instrument (Sections 6.2.2, 6.2.3) and during the instrument correlation with the MAAP in section 6.2.4 when measuring coarse mode (i.e., PM10) particles. While these discussions are important, the authors need to qualify these discussions with the facts that neither the CAPS nor the MAAP have been extensively tested with coarse mode particles (to my knowledge). For example, the discussion of errors in the MAAP on page 16 mention the RAOS study and the Mueller et al. (2011a) study, but neither of these studies included coarse mode particles (i.e., particles greater than 1 micron in diameter). Thus, while including these discussions here is important and relevant to the ambient results, the authors need to acknowledge that neither technique has been extensively studied for coarse mode particles (to my knowledge), which limits the quantitative aspect of these discussions. Furthermore, a large fraction of the discussions on Figure 8 and the subsequent analysis relies on the coarse-mode and full dataset (which is highly biased by a coarse-mode). The authors should at least

qualify these results with respect to the lack of coarse mode quantification in both instrument techniques.

This is a good point and we generally agree: also to our knowledge there has been no extensive and dedicated studies on the ability of the MAAP to measure absorption coefficients for super-micrometer aerosols. We do not think this point affects the majority of the discussion in Sect. 6.2, which is focused on how the CAPS PMssa measurements responds to the presence of aerosols with strongly forward-focused scattering phase functions. However, we agree that it is an important qualification to highlight. We will add explicit statements to both Sections 3.1.1 and 6.2 to highlight this issue.

15.) Page 25 line 765, very nice correlation for high time resolution data. What does the correlation plot (or histogram of the ratios) of CAPS PMssa b_abs to SP2_rBC look like for the data presented in Figure 7? Since you explicitly state not to look at the quantitative value, I did that. The apparent ratio is 8 m2/g, which is very high for a wavelength of 780nm. What are the SSA values for these samples? The authors point out that the one unknown (i.e., not measured) component during this study is the scattering truncation (i.e., size distributions). For scattering truncation uncertainties, the SSA would have to be high to attempt to account for these apparently high MAC values. What were the size distributions of rBC measured by the SP2? What about the scattering measurements by SP2? Could the SP2 be saturating at these high rBC levels? What about absorption enhancement due to coated rBC particles (would be ~1.5x assuming a 1/wavelength from 550nm 7.5 m2/g MAC, which might not be unreasonable)? Perhaps, it would be either better to (a) remove one or both axes if considered non-quantitative or (b) provide a potential rationale for this apparent issue.

As well as the uncertain truncation correction constant the other main quantitative issue in this figure is related to the relatively small size of the freshly-emitted rBC cores, which means the SP2 was unable to detect an unknown fraction of the total rBC mass (this issue is discussed at length by Pileci et al. 2020, including with reference to this particular field campaign). Underestimated total rBC mass from the SP2 is the most likely reason for the high apparent MAC value. In combination with the unknown truncation correction constant, this is also the reason we don't think there is any value in analyzing or constraining the quantitative aspects of this result any further. However, we agree it is important to add further rationale about this for readers who are as curious as the reviewer. In the revised manuscript we will add statements in the main text and in the caption of Fig. 7 that the absolute rBC mass concentrations should also be treated with caution, since the SP2 was unable to detect all of the rBC mass present due to the small size of the rBC cores.

Pileci, R. E., Modini, R. L., Bertò, M., Yuan, J., Corbin, J. C., Marinoni, A., Henzing, B. J., Moerman, M. M., Putaud, J. P., Spindler, G., Wehner, B., Müller, T., Tuch, T., Trentini, A., Zanatta, M., Baltensperger, U. and Gysel-Beer, M.: Comparison of co-located rBC and EC mass concentration measurements during field campaigns at several European sites, Atmospheric Measurement Techniques Discussions, 1–32, doi:https://doi.org/10.5194/amt-2020-192, 2020.

16.) Section 6.2. This section analyzes a comparison of the CAPS absorption measurements compared with the MAAP absorption measurements. As noted above in points 2 and 14, both instruments have

associated errors for submicron and potential unknown errors for supermicron. The authors state (page 25, line 784) that this was done to "perform a full quantitative assessment of the ability of the CAPS PMssa to measure absolute aerosol absorption coefficients…". In reality, this section is a direct instrument to instrument comparison, rather than a quantitative assessment for the measurement of absolute aerosol absorption coefficients. It is likely that the errors in the MAAP are smaller than the errors in the CAPS b_abs, but they should still be included. What MAC value was assumed for the MAAP to derive the b_abs values? On page 28, when discussing the ~20% discrepancy between the CAPS b_abs and MAAP b_abs, the only errors discussed are those of the CAPS geometry correction factor and truncation factor, which are only on the order of 2-9%. The MAAP uncertainties stated on page 16 indicate similar orders of magnitude (5-7%), but are not discussed.

Again, this is a good point and we generally agree. We will adopt the language accordingly, including:

- Rewording the phrase "…a full quantitative assessment of the ability of the CAPS PMssa to measure absolute absorption coefficients" to something like: "a direct instrument to instrument comparison in order to gain insight into the ability of the CAPS PMssa to measure absolute absorption coefficients".
- Mention of MAAP-related uncertainties in Section 6.2 (specifically in the discussion on page 28 as suggested). We note that we refrained from plotting MAAP-related error bars in Fig. 8 because they are too small and overly complicate an already busy plot. However, we will add a statement to the caption of Fig. 8 that MAAP-related uncertainties are not displayed for visual clarity.

The manufacturer-specified MAC value of 6.6 m2/g was used to convert the eBC concentrations reported in the output of the MAAP firmware to absorption coefficients. We will include this information in Section 3.1.1

17.) Figure 8. This figure shows a lot of useful information tightly packed into one figure. The problems of using log-log plots for instrument intercomparisons is that it (a) blows up the noise at low signals and (b) removes any potential negative values from the comparisons. My first question here is how were the fits done for this figure? For example, the fit in the center plot does not go through the data shown at all. Thus, it is not a linear fit of the data in linear space and placed on log-log axes. Further, the fits assume that there is no y-intercept (or x-intercept). If it is a power law fit to the data, then there must be a lot of data not shown that is highly biasing the power fit. How do the fits account for the negative values? More information is warranted on the fitting approach and why it was chosen.

The decision to display these results on a log-log scale was a deliberate one. We believe it is important to show the level of agreement across the full range of measurements. In contrast, display on linear axes would visually preference those measurements performed at the highest b_abs values. Although these particular scatterplots don't show much evidence of random noise at low signal levels for the given averaging level of 1 hour, we believe that even if they did this would rather be an argument for showing the results on a log-log scale, i.e., we aim for maximum transparency, and prefer that readers have the opportunity to judge for themselves at what signal levels noise starts to become an issue.

The major drawback with our choice is that, as suggested, the negative CAPS PMssa measurements cannot be displayed. In this instance however, we believe the most pertinent piece of information is that negative measurements were sometimes present, not how those negative measurements actually looked like. Therefore, we added text boxes indicating the fraction of the CAPS PMssa measurements that were negative in order to get around this issue.

For these reasons, we prefer to keep these measurements displayed as they currently are on log-log axes.

18.) Figure S15 caption indicates that the fit lines in Figure 8 are not fits, but rather mean ratios…. – this information needs to get into the main text and main figure captions!!! By taking the mean ratio, the resulting values are heavily influenced by noise in the smaller measurements. This would explain why the orange lines do not pass through the majority of the data points, especially at higher values of b_abs and lower values of SSA, where one would expect better agreement. Another potential approach that could be taken would be to estimate a conservative limit (for example 50 Mm-1 was chosen for this purpose when looking at the geometry correction factors above) and then look at the histograms of the ratios (again similar to the geometry correction factor analysis). This approach might better define the mean, variance, and skewness of the resulting histograms, where the skewness could be used as a measure of how well the truncation models are being applied, rather than the vague (though true) color trends.

Firstly, we apologize for the omission of details concerning the mean ratio calculation in the original submission. This will be corrected in the revised manuscript with appropriate description added to both Sect. 6.2 and the caption of Fig. 8.

The decision to display lines with gradient of mean ratio rather than standard linear fits was taken to avoid biasing the fits towards the measurements at high b_abs values. One key difference between the measurements shown in Fig. 8 and those used to calculate the cross calibration constants (where, as suggested, a conservative limit of 50 Mm-1 was used, e.g. Figs. S3, S4, S10 and S11), is that the measurements in Fig. 8 are hourly averages, not 1 second measurements, and therefore they are less affected by noise at very low signal levels (a good example of how averaging to 1 hour removes the random measurement noise at low signal levels is shown in Fig. S9). Therefore, noise at low signal levels does not bias the mean ratios displayed in Fig. 8 substantially.

The fact that the lines of mean ratio do not pass through the majority of the points plotted in Fig. 8e is rather to do with the fact that this subplot contains a sizeable fractions of negative CAPS PMssa measurements that are not displayed on the log-log axes. These negative values are the result of over-corrected truncation, not noise at low signal levels, as we discuss at length in Section 6.2.

Finally, we experimented considerably with more quantitative measurements for assessing the performance of different truncation approaches (e.g. histogram skewness, bias relative to the MAAP as a function of SSA). However, our goal is not optimization of the parameters in our Mie-based truncation correction model. I.e., we're not looking to identify a single best set of parameters for this particular dataset. The reason is that this optimized set of parameters would not be generally applicable to other studies, so there would be little value in reporting it. Rather, our aim is to show the sensitivity of the

CAPS PMssa b_abs measurements to different truncation correction scenarios, to provide general insight and intuition into the effects of the correction on the final results. For this reason we think it is more appropriate to rely on qualitative metrics such as the color change rather than specific quantitative metrics.

19.) Equation A11 takes the probably, R, which is an average of the s and p polarizations (see Eq A9), and takes it to the power of the number of estimated reflections for a light ray to exit the glass tube. As s and p polarizations have significantly different reflection probabilities as a function of incident angle, equation A11 is an approximation. Ideally, one would calculate R_s and R_p separately for the two polarizations and then average. Not sure how much of a difference this would make, but it would be more accurate.

We tested this by calculating light collection efficiency function curves with the reflection probabilities for the different polarization states averaged before and after the raising to the power of the number of reflections. The result is displayed in the following figure. It is seen that averaging before or after has no noticeable different on the calculated efficiency curves (the 2 curves overlap very closely). Therefore, we decided to make no changes to Eq. (A11).

[Figure]

**Referee #2: Anonymous**

**General comments:**

This study provides a detailed characterization and thorough discussion on the performance of commercialized CAPS PMssa in determining aerosol light absorption using the extinction minus scattering method and the associated uncertainties with this method. While the paper is well written in many aspects, I am a bit concerned that the key points and many useful information may be missed by the readers because of the extraordinary length of the manuscript. In general, I feel that Section 2 and 3 can be combined and shortened to make the experimental method clearer and to the point. Specifically, in section 2.1 and 2.2, Onasch (2015) was heavily cited and many of the description of the instrument is repetitive. Section 2.3 and the sub-sections of Sect 2.2 are also mixed with extensive discussion, which can either to be moved to later sections or to Appendix. Also, the results from the Bologna campaign

seems less relevant to the main focus of this study: identifying and quantifying uncertainties in the absorption measurement due to the scattered light truncation effect and cross calibration constant. I recommend publication of this manuscript after the above points are considered.

We thank the reviewer for his/her review and comments regarding the manuscript length. While we acknowledge that the manuscript is lengthy, we believe that this is necessary since it is only by considering all the specific aspects of the instrument and data processing that one is able to fully understand the experimental results displayed in Sections 4, 5 and 6.

Section 2.1 indeed contains some instrument details that are already reported by Onasch et al., (2015). However, we have endeavored to restrict this information to only those details that are pertinent to understanding this study. Since this Section only consists of 4 short paragraphs, we believe there is not much to be gained by further cutting it down. In contrast, retaining these details makes the manuscript self-contained and prevents readers from constantly having to switch between 2 different papers.

Regarding Sections 2.2. and 2.3, most of these details have not been previously discussed at length before in the literature and, as mentioned, they are necessary to understand the experimental results that follow. Therefore, we believe the extensive discussion in these sections is necessary.

Finally, although the Bologna result cannot be interpreted as quantitatively as the rest of the results we believe they are extremely important to show. They demonstrate a key advantage of the CAPS PMssa relative to filter-based absorption photometers: the ability to measure at very high time resolution. This key advantage has not been demonstrated extensively in the literature yet, which is why we believe the results are pertinent and worth including in this paper.